# Limited overlap between genetic effects on disease susceptibility and disease survival

Understanding disease progression is of high biological and clinical interest. Unlike disease susceptibility, whose genetic basis has been abundantly studied, less is known about the genetics of disease progression and its overlap with disease susceptibility. Considering nine common diseases ($n_{cases}$ ranging from 11,980 to 124,682) across seven biobanks, we systematically compared genetic architectures of susceptibility and progression, defined as disease-specific mortality. We identified only one locus substantially associated with disease-specific mortality and showed that, at a similar sample size, more genome-wide significant loci can be identified in a genome-wide association study of disease susceptibility. Variants substantially affecting disease susceptibility were weakly or not associated with disease-specific mortality. Moreover, susceptibility polygenic scores (PGSs) were weak predictors of disease-specific mortality, while a PGS for general lifespan was substantially associated with disease-specific mortality for seven of nine diseases. We explored alternative definitions of disease progression and found that genetic signals for macrovascular complications in type 2 diabetes overlap with similar phenotypes in the general population; however, these effects are attenuated. Overall, our findings indicate limited similarity in genetic effects between disease susceptibility and disease-specific mortality, suggesting that larger sample sizes, different measures of progression, or the integration of related phenotypes from the general population is needed to identify the genetic underpinnings of disease progression.

Genome-wide association studies (GWASs) have been successful in uncovering the genetic basis of human diseases by using a relatively straightforward study design that compares individuals with the disease to controls[1–3]. This approach is well-suited to identify loci associated with disease susceptibility, but it remains unclear whether these results can also inform on the biology of disease progression. Studying the genetic basis of disease progression is relevant for at least two reasons. First, biological insights from the study of disease progression can be more relevant for drug target discovery since many medicines are developed to cure a disease rather than prevent its occurrence. Second, most individuals approach the healthcare system once they develop a disease or its symptoms, and predicting disease progression is, in most diseases, an important clinical challenge.

In the past years, several GWASs of disease progression have been performed (see Supplementary Table 1 for a detailed review), but the number of progression-specific loci discovered has been limited.

In cancer, GWASs have primarily focused on disease survival and have generally been unsuccessful in identifying genome-wide significant signals. For example, a GWAS of breast cancer survival in over 96,000 patients did not identify any robust association[4] and failed to replicate two loci found in the previous largest GWAS of breast cancer survival[5]. Among neurological conditions, GWASs have focused on disease survival as well as cognitive or motor decline. In one of the largest studies, researchers identified three new loci associated with the progression of Parkinson's disease[6]. A recent study on multiple sclerosis progression identified a locus indicating involvement of

✉e-mail: andrea.ganna@helsinki.fi

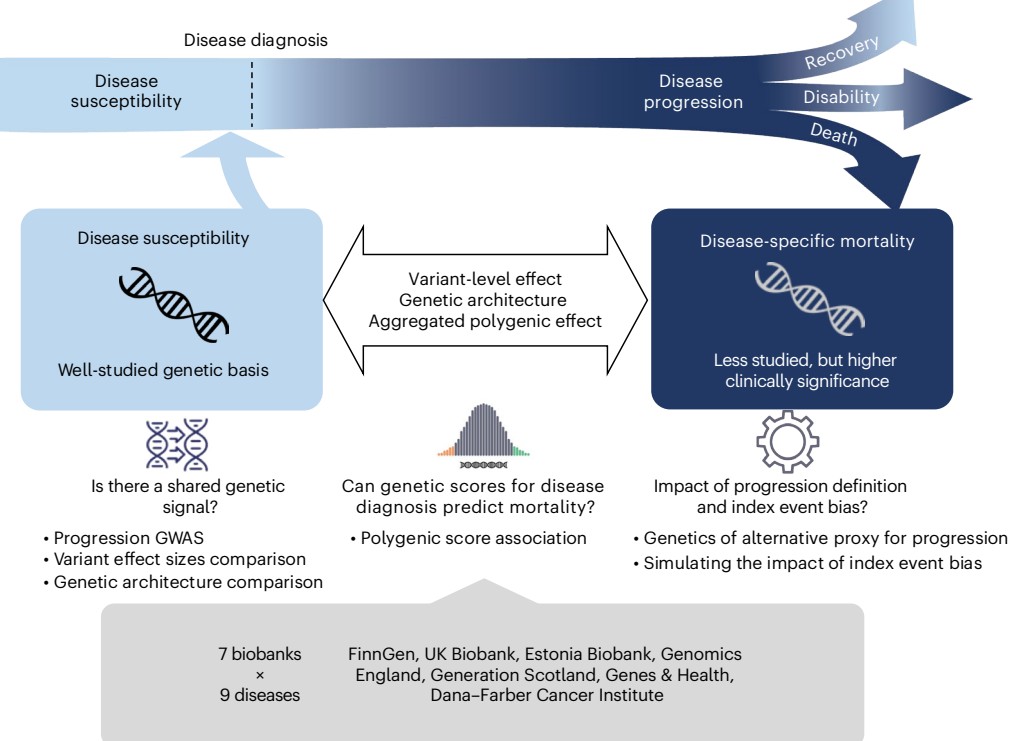

**Fig. 1 | Study overview.** Using data from seven biobanks, we investigated the genetic similarity between disease susceptibility and disease progression, defined as disease-specific mortality. We selected nine diseases and ran GWASs of disease-specific mortality among disease individuals. We then compared the genetic architecture of disease susceptibility and mortality, focusing on both single variant and aggregated polygenic effects. We further explored the impact of alternative progression definitions and the theoretical impact of index event bias on the results.

the central nervous system in disease outcome, as opposed to the enrichment of immunological-related signals observed for disease susceptibility[7]. However, it is worth noting that older studies of multiple sclerosis outcomes have failed to replicate in larger ones[8,9]. In cardiovascular diseases, studies have focused on disease recurrence, and initial results from the Genetics of Subsequent Coronary Heart Disease (GENIUS-CHD) consortium showed the strongest GWAS signal for coronary artery disease was not associated with subsequent events[10]. In Crohn's disease, a study identified four loci associated with disease progression, indicating a distinct genetic contribution to disease susceptibility[11].

Apart from single-variant-level effects, some studies examined the aggregate effect of many genetic variants. Most of them suggested that polygenic scores (PGSs) for disease susceptibility do not transfer well to disease progression[11–13], although they might outperform other disease-specific biomarkers in the case of cardiovascular diseases[14].

Some authors have highlighted the challenges in interpreting results from genetic studies of disease progression due to the bias induced when individuals are selected based on their disease status. If common causes of susceptibility and progression are not accounted for, association results can be unreliable due to what is called an index event bias[15], and several approaches to detect and correct for index event bias have been proposed[16,17].

Large-scale biobanks linked with longitudinal electronic health records have accelerated research into the genetic basis of disease progression and provide sufficient sample size to answer the following two key questions: (1) do genetic predictors that influence disease susceptibility have a similar impact on disease progression? (2) Can we use PGSs for disease susceptibility to predict disease progression? In this study, we aim to provide empirical answers to these two questions by focusing on a specific but commonly used definition of disease progression: disease-specific mortality (Fig. 1). Through an international collaboration across multiple large-scale biobanks, we systematically compared genetic architecture of disease susceptibility and mortality for nine common diseases, focusing on both single variant and aggregated polygenic effects.

## Results

### Participating biobanks and disease of interest

We considered nine common diseases that substantially increase mortality risk in the general population and have a large public-health impact (Table 1). We confirmed disease association with mortality using nation-wide Finnish data and observed a hazard ratio (HR) for 20-year mortality ranging from 1.31 for type 2 diabetes to 3.61 for chronic kidney disease in females[18] (Supplementary Table 2). We identified diseased individuals based on consistent disease definitions captured via electronic health records or registry data across eight longitudinal studies—FinnGen[19], UK Biobank[20], Estonian Biobank[21], Generation Scotland[22], Genomics England[23], Genes & Health[24], Dana–Farber Cancer Institute[25] and BioMe. The number of individuals included ranged from 124,682 individuals with type 2 diabetes to 11,980 individuals with Alzheimer's disease (Table 1). All individuals with a disease were followed up for at least three months. We defined disease-specific mortality based on death certificates in which the disease of interest was listed as the primary or secondary cause of death. One participating biobank did not have information on causes of death, and we used overall death instead (Supplementary Table 2). We observed the highest cause-specific mortality rate for Alzheimer's disease (40%) in FinnGen and the lowest for type 2 diabetes (3%) in the Estonian Biobank.

### Susceptibility SNPs do not affect disease-specific mortality

For each disease, we carried out a GWAS of disease-specific mortality among disease individuals using Cox proportional hazard model as implemented in GATE[26] or SPACox[27] (Supplementary Figs. 1–9).

**Table 1 | Total sample sizes for disease-specific mortality GWAS for each disease and percentage of mortality by year**

| Disease | Sample size | | Percentage of deaths within 2 years | Percentage of deaths within 5 years | Percentage of deaths within 10 years |
| --- | --- | --- | --- | --- | --- |
| | Disease-specific deaths (n) | Diseased individuals (n) | | | |
| Prostate cancer | 3,496 | 31,668 | 2.89% | 6.14% | 8.99% |
| Breast cancer | 3,226 | 39,750 | 1.61% | 4.32% | 6.36% |
| Colorectal cancer | 4,051 | 20,434 | 9.48% | 16.59% | 19.11% |
| Coronary artery disease | 12,661 | 97,849 | 1.82% | 4.00% | 7.06% |
| Type 2 diabetes | 6,372 | 124,682 | 0.42% | 1.15% | 2.51% |
| Chronic kidney disease | 1,973 | 32,757 | 2.03% | 4.07% | 5.61% |
| Alzheimer's disease | 4,352 | 11,980 | 9.00% | 21.98% | 33.31% |
| Heart failure | 7,902 | 102,063 | 1.70% | 2.85% | 3.98% |
| Stroke | 2,037 | 41,484 | 1.25% | 2.90% | 5.08% |

See also Supplementary Table 2 for details.

On top of all common GWAS covariates, all analyses were also adjusted for age at disease diagnosis for the following two reasons: (1) age is a strong predictor of mortality; (2) age of onset has a nontrivial genetic contribution partially overlapping with disease susceptibility[28], and we are instead interested in genetic effects on disease-specific mortality.

Of all nine diseases studied, we identified only one locus associated with disease-specific mortality at genome-wide significance ($P < 5 \times 10^{-8}$). The locus (rs7360523) on chromosome 20, close to *SULF2*, was associated with disease-specific mortality among patients with heart failure.

We asked whether well-established signals for disease susceptibility were associated with disease-specific mortality (Fig. 2 and Supplementary Table 3). For each disease, we compared the effect sizes from the largest published GWAS with the results from our GWAS of disease-specific mortality. In total, 804 lead variants were reported from all susceptibility GWASs. None of them were substantially associated with disease-specific mortality after multiple testing correction ($P < 0.05/804 = 6.22 \times 10^{-5}$), while 392 showed the same effect direction, which is no more than expected by chance (probability of observing same direction of effect direction = 0.49 (95% confidence interval (CI) = 0.45–0.52); binomial test against 0.5, $P = 0.5028$).

The only disease-specific mortality locus identified for heart failure also did not show comparable effect on heart failure susceptibility ($P = 0.87$ in susceptibility GWAS with opposite direction of effect). The low number of genome-wide signals for disease-specific mortality was consistent with the lower estimated heritability compared to the GWAS of disease susceptibility (Supplementary Table 2).

**Sample sizes do not explain low heritability of mortality**

To assess whether the overall lack of substantial genetic signals for disease-specific mortality was simply due to smaller sample sizes compared to the GWASs of disease susceptibility, we performed a down-sampling experiment in FinnGen and UK Biobank by imposing the same effective sample size for both analyses. To further make the two analyses comparable, the GWASs of disease susceptibility were conducted using survival analysis with age as time scale and disease diagnosis as outcome. The GWASs of disease susceptibility returned 33 genome-wide significant loci across five of the nine tested diseases, while the GWASs of disease-specific mortality returned no genome-wide significant results (Table 2).

**Susceptibility PGSs are weak predictors of mortality**

We investigated the joint effects of genetic variants associated with disease susceptibility in predicting disease-specific mortality. For each disease, we constructed a PGS using results from the largest GWAS of disease susceptibility. All the PGSs were strongly associated with disease susceptibility. The HRs for 1 s.d. in the PGS ranged from 1.17 (1.15–1.18) for stroke to 1.90 (1.88–1.93) for prostate cancer (dashed line in Fig. 3). On the contrary, the same PGSs were weakly or not associated with disease-specific mortality (orange dots in Fig. 3). For example, although strongly associated with disease susceptibility, a PGS for breast cancer showed no association with breast cancer mortality (HR = 0.98 (0.93–1.02)). The most substantial association was observed between the heart failure PGS and heart failure mortality (HR = 1.09 (1.07–1.12)), while the PGSs for chronic kidney disease and prostate cancer trend towards having a protective effect on mortality (HR = 0.96 (0.91–1.02) and HR = 0.96 (0.93–1.00), respectively).

To assess the robustness of these results, we conducted a range of sensitivity analyses. First, we assessed whether using a less specific definition of disease progression, namely all-cause mortality, would impact the observed results. We observed significantly larger correlation coefficients of susceptibility PGSs on disease-specific mortality than on all-cause mortality in five of nine diseases (Supplementary Fig. 10 and Supplementary Table 7). Second, we only considered individuals who developed the disease after study enrollment (Supplementary Fig. 11a and Supplementary Table 8) as a way to account for survival bias, which might explain some of the negative associations between PGSs and cause-specific mortality. Nonetheless, results were consistent (correlation coefficient $r$ between effect sizes $\beta$ in the main analysis and sensitivity analysis = 0.96), and we continued to observe a negative association between a PGS for prostate cancer and prostate cancer mortality. Third, we considered different maximum follow-up lengths (2, 5 and 10 years) because we reasoned that deaths occurring shortly after disease diagnosis were more likely to be caused by the disease. However, results were overall comparable across follow-up lengths (correlation coefficient $r$ between effect sizes in main analysis and sensitivity analysis = 0.67, 0.78 and 0.92 for 2, 5 and 10 years, respectively; Supplementary Fig. 11b and Supplementary Table 8), and contrary to our expectation, some diseases (for example, heart failure) showed a stronger association between the susceptibility PGS and disease-specific mortality when considering longer rather than shorter follow-up lengths (effect size $\beta = -7.99 \times 10^{-4}$, 0.03 and 0.05 for 2, 5 and 10 years, respectively). Fourth, we evaluated whether adjusting the analyses for age at diagnosis could mask an age-specific effect of PGS on cause-specific mortality, for example, because such effect was only observed among young or old patients. We observed the largest change in $z$ score for PGS effect on disease-specific mortality between lower and upper 50% quantile diagnosed age groups only for Alzheimer's disease ($\Delta z = 3.51$; Supplementary Fig. 12 and Supplementary Table 9). That is, the association between Alzheimer's disease PGS and mortality was substantial only among younger but not older patients. Finally, we tested the effect of using only unrelated individuals in FinnGen and found the result to be robust (Supplementary Fig. 13 and Supplementary Table 7). We also carried out the same analyses

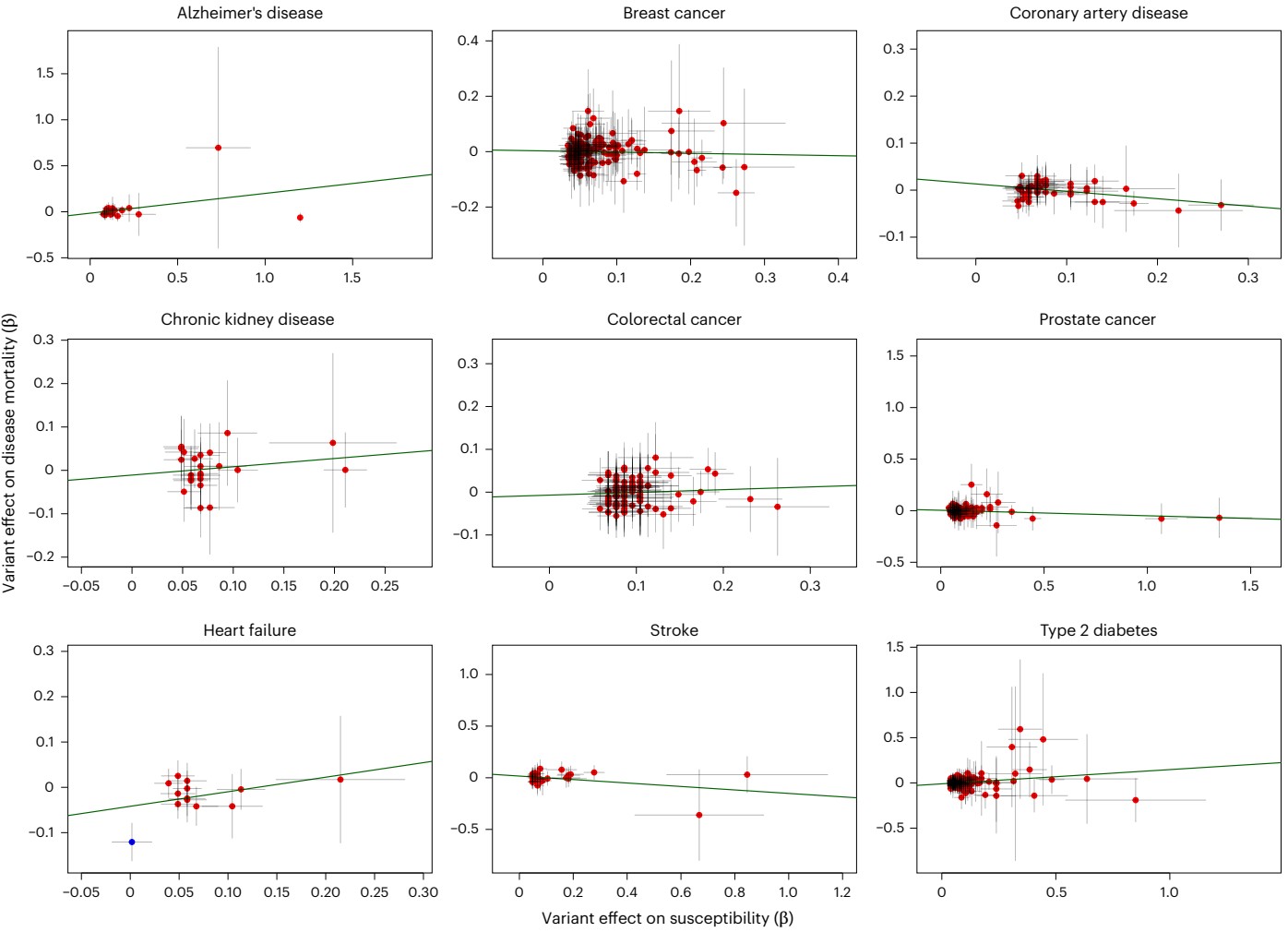

**Fig. 2 | Relationship between variant effects (one for each locus) on disease susceptibility (x axis) and disease-specific mortality (y axis).** Variants were selected either because of genome-wide significance for susceptibility in the largest disease-specific GWAS or because of genome-wide significance for disease-specific mortality in the current study, indicated by color of the dot (red, significant in susceptibility GWAS; blue, significant in mortality GWAS). Data are presented as 95% CI for GWAS effect sizes ($\beta \pm 1.96$ s.e. from corresponding GWAS). See also Supplementary Table 3 for quantitative results and sources of susceptibility GWAS summary statistics. Only one locus for heart failure mortality was genome-wide significant across nine mortality GWASs.

using non-European individuals from Genes & Health. However, due to limited power, no conclusion could be drawn (Supplementary Fig. 15). Forest plots of effects from each participating European biobank are shown in Supplementary Fig. 14.

**A general longevity PGS can predict mortality better**

Having established that susceptibility PGSs are weakly associated with disease-specific mortality, we reasoned that other PGSs that are better proxies of disease-specific mortality could show stronger associations. First, we considered PGSs constructed directly from our GWASs of disease-specific mortality. For diseases where power allowed, we derived PGSs using weights from the meta-analyzed GWAS results from all biobanks except for FinnGen and tested the association between PGS and disease-specific mortality within FinnGen. Surprisingly, of all four diseases, only colorectal has its PGS nominally associated with disease-specific mortality, with a rather marginal effect ($\beta = 0.0523$, $P = 3.58 \times 10^{-2}$; Supplementary Fig. 17 and Supplementary Table 10).

Second, we considered a PGS for general longevity derived from the largest lifespan GWAS[29] under the assumption that it might capture some of the genetic effects related to disease survival. The longevity PGS was substantially associated with disease-specific mortality for seven of nine diseases ($P < 0.005$, accounting for the number of diseases tested), and for seven diseases, it shows larger HR than a PGS for

susceptibility (Fig. 4 and Supplementary Table 5), including the three diseases where we observed protective effects of susceptibility PGSs.

We also tested another composite mortality PGS developed from the genetics of multiple risk factors[30] in FinnGen. The composite mortality PGS was substantially associated with disease-specific mortality for four of nine diseases, and outperformed longevity PGS in predicting type 2 diabetes and coronary artery disease mortality (Supplementary Fig. 16 and Supplementary Table 12).

**Alternative progression definitions for type 2 diabetes**

Disease-specific mortality is widely accessible across biobanks but can be an inaccurate proxy of progression for some diseases. For type 2 diabetes, macrovascular complications (defined as coronary artery disease, stroke or peripheral arterial disease) and microvascular complications (defined as diabetic retinopathy, nephropathy, and neuropathy) are more clinically relevant measures of disease progression that are captured in electronic health records. We ran time-to-event progression GWAS among type 2 diabetes patients for both macro ($n_{cases} = 17,332$ of 85,188 eligible type 2 diabetes patients) and microvascular ($n_{cases} = 5,798$ of 103,185 eligible type 2 diabetes patients) complications in FinnGen and UK Biobank, and meta-analyzed the results. We identified one genome-wide significant locus on chromosome 9 for macrovascular complications, but none for microvascular

**Table 2 | GWAS power comparison between disease-specific mortality and disease susceptibility under the same sample size and GWAS model in FinnGen and UK Biobank**

| Disease | Number of deaths ($n$) | Censored individuals ($n$) | Number of GWAS loci | |
|---|---|---|---|---|
| | | | Disease-specific mortality | Down-sampled susceptibility |
| Prostate cancer | 3,074 | 24,982 | 0 | 17 |
| Breast cancer | 1,924 | 30,181 | 0 | 1 |
| Colorectal cancer | 2,727 | 11,906 | 0 | 0 |
| Coronary artery (heart) disease | 11,088 | 64,563 | 0 | 5 |
| Type 2 diabetes | 5,655 | 97,632 | 0 | 5 |
| Chronic kidney disease | 1,505 | 24,853 | 0 | 0 |
| Alzheimer's disease | 4,352 | 7,628 | 0 | 5 |
| Heart failure | 5,348 | 69,997 | 0 | 0 |
| Stroke | 1,783 | 35,830 | 0 | 0 |

In the table, we present numbers of independently associated genome-wide significant loci from disease-specific mortality GWAS and down-sampled disease susceptibility GWAS. We report no significant loci for heart failure, in contrast to what is reported in Fig. 2, because these GWAS analyses were conducted only in FinnGen and UK Biobank.

complications (Fig. 5 and Supplementary Fig. 18). The locus was not associated with type 2 diabetes susceptibility ($P = 0.2898$ from UK Biobank and FinnGen meta-analysis), but it was associated with type 2 diabetes mortality ($P = 5.6 \times 10^{-4}$), highlighting how disease-specific mortality can capture, albeit imprecisely, disease progression. We wondered whether this genetic signal was shared with cardiovascular conditions in individuals without type 2 diabetes and performed a GWAS for age of onset for coronary artery disease, stroke, or peripheral arterial disease in the UK Biobank and FinnGen. For this experiment, we performed random down-sampling to match the available sample sizes for the progression GWAS. We observed a strong overlap in the signals from the GWAS of diabetic macrovascular complications and the GWAS of cardiovascular conditions in individuals without type 2 diabetes (Fig. 5). In other words, the locus detected for type 2 diabetes macrovascular complications was also detected by a GWAS of similar phenotypes in individuals without type 2 diabetes. However, the effect was overall reduced among individuals with type 2 diabetes (leading SNP $-\beta = 0.09$, $P = 3.62 \times 10^{-16}$ versus $\beta = 0.11$, $P = 1.36 \times 10^{-23}$).

As we have previously shown that a longevity PGS created in the general population was predictive of disease-specific mortality, we wondered whether a similar observation would hold for these refined definitions of type 2 diabetes progression. To test this, we considered PGSs for coronary artery disease[31] and stroke[32] as predictors of type 2 diabetes macrovascular complication, and PGSs for age-related macular degeneration[33] and chronic kidney disease[34] as predictors of type 2 diabetes macrovascular complication.

A PGS for coronary artery disease was a stronger predictor of macrovascular complications among type 2 diabetes patients than the PGS of type 2 diabetes susceptibility (HR = 1.19 (1.17–1.21) versus HR = 1.04 (1.03–1.06)). For microvascular complications, we did observe a nominally significant association for the PGS of age-related macular degeneration (HR = 1.03 (1.01–1.06)), but not chronic kidney disease (HR = 1.00 (0.98–1.03); Supplementary Fig. 19 and Supplementary Table 6).

**Potential role of index event bias**

Noticing an attenuation of variant effects in progression GWASs with the same sample sizes, we suspected that index event bias could have a role. Therefore, we carried out simulation under a simple liability threshold model and explored the impact of index event bias by introducing a shared nongenetic risk factor accounting for various proportions of the liability in disease susceptibility and progression. We compared the simulated effect of causal genetic variants on progression with the observed effect from the progression GWASs and found larger differences when the shared nongenetic component accounted for higher liability variance, indicating higher impact of index event bias (Supplementary Fig. 20). A correction approach similar to slope-hunter[16] reduced the bias, improving the concordance with the true simulated effects. However, in the scenario of a low progression heritability, which is consistent with our empirical findings for disease mortality, index event bias correction showed limited impact as we observed no genetic variants substantially associated with disease progression before or after bias correction (Supplementary Table 11).

## Discussion

In this study, we systematically explored the overlap of genetic effects on disease susceptibility and a common measure of disease progression, disease-specific mortality, for nine common diseases. By conducting the largest within-patient GWAS of disease-specific mortality to date, we found that (1) lead variants affecting disease susceptibility do not have comparable effect sizes on disease mortality; rather, they show little effect and no significant association with disease-specific mortality in GWAS; (2) at a similar sample size, GWASs of disease-specific mortality identified fewer genome-wide significant loci than GWASs of disease susceptibility, suggesting that GWASs of disease progression might require larger sample size or more refined phenotypes than GWAS of disease susceptibility; (3) disease susceptibility PGSs do not transfer well on disease-specific mortality, suggesting that current PGSs are more suitable for identify individuals at high risk of developing a disease rather than those more likely to suffer from the worst consequences and (4) if available, traits measured in a general population, but related to disease progression, can inform the genetic underpinning of disease progression, and PGSs derived from these related traits can predict progression better than PGSs for disease susceptibility.

The limited overlap between genetic effects on disease susceptibility and disease progression may have several explanations.

First, genetic influences on disease progression might be too small to detect. External environmental effects such as treatment choice, treatment response, quality and access to care might have a disproportionate impact on disease progression as compared to disease susceptibility, thus limiting the genetic influence. Heterogeneity in patients and their treatments has a substantial role in disease progression, and we are currently unable to account for all this heterogeneity. Using data from clinical trials rather than observational studies and including finer measurements, such as disease-relevant biomarkers, could obviate these shortcomings. We also noticed that adjusting for age at disease diagnosis reduces the overlap between susceptibility and progression because variants that increase disease susceptibility are often associated with earlier disease diagnosis[28]. Previous studies have demonstrated impact of adjusting for age in disease progression analyses[35] and suggested that association between PGSs and measurement of disease progression may be mediated by age.

Second, our definition of disease progression may be a poor proxy for the biological mechanisms that impact disease progression. Our approach aims to compare progression across multiple diseases, but this comes at the expense of a tailored definition of progression for each disease. Nonetheless, disease-specific mortality has been widely used as a measure of progression[6,36–38]. Association between composite genetic score of various risk factors, including several diseases considered in our study, and all-cause mortality has also been reported[30]. We tested this score in FinnGen and observed better prediction performance for some diseases of interest. In practice, biobank-based studies of disease progression often require simplification of definitions to maximize sample size, and disease-specific mortality information is

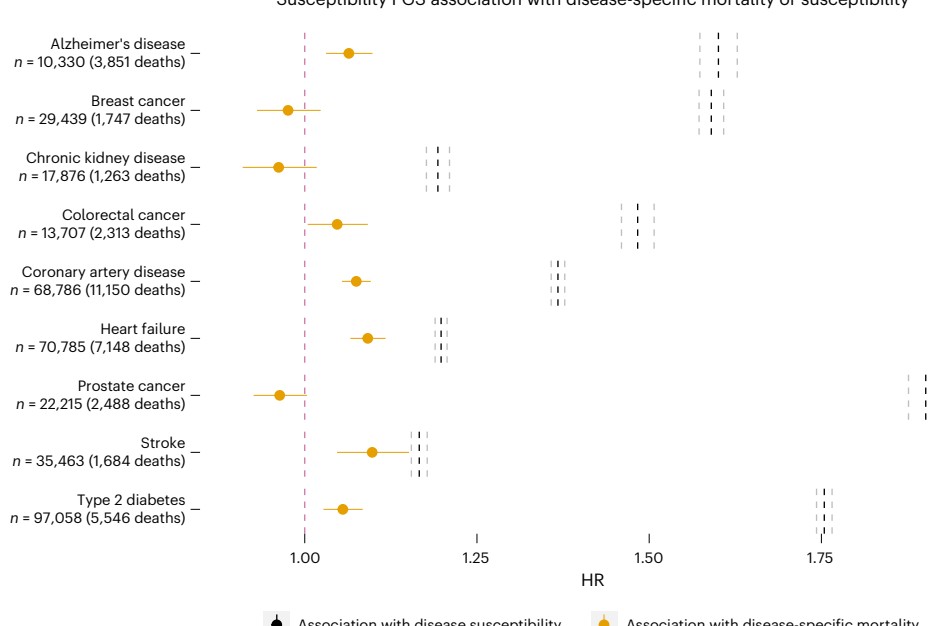

**Fig. 3 | Association between PGS for disease susceptibility and either disease-specific mortality (orange dot) or susceptibility (dashed line).** Disease susceptibility PGSs were derived from published large-scale GWAS for each disease. PGS associations with both disease susceptibility and disease-specific mortality were examined using a Cox proportional hazards model. The sample size reported on the y axis refers to the disease-specific mortality analyses, and the sample size for association with disease susceptibility can be found in Supplementary Table 4. Horizontal solid lines represent 95% CI for PGS association with disease-specific mortality HR ($\exp(\beta \pm 1.96$ s.e.)). The vertical dashed lines in black and gray represent association with disease susceptibility HR and 95% CI, respectively, ($\exp(\beta \pm 1.96$ s.e.)).

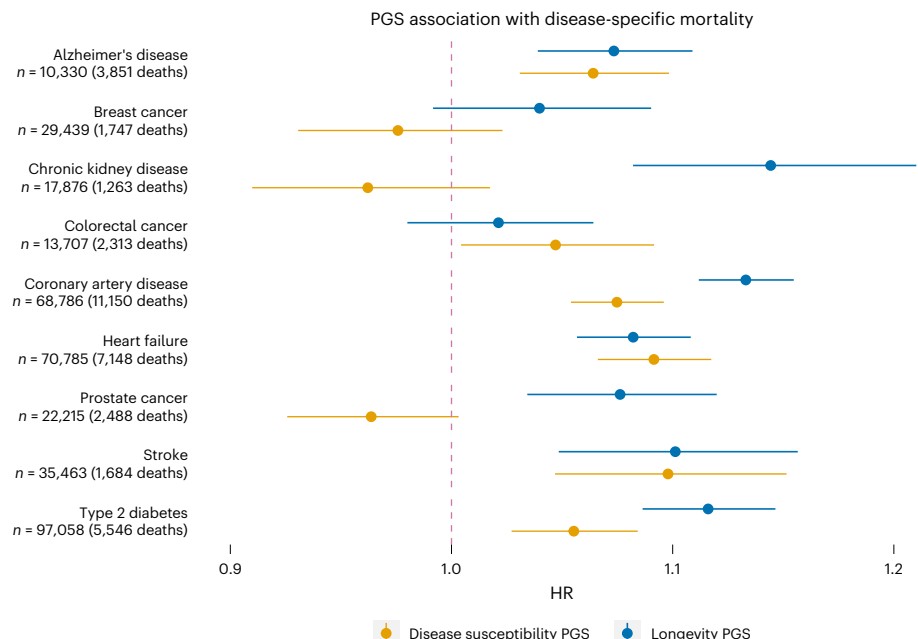

**Fig. 4 | Association between a PGS for disease susceptibility (orange dots) and longevity (blue dots) with disease-specific mortality.** Disease susceptibility PGSs were derived from published large-scale GWAS for each disease. Longevity PGS was derived from ref. 29. Horizontal solid lines represent 95% CI for PGS association with disease-specific mortality HR ($\exp(\beta \pm 1.96$ s.e.)). For quantitative results, see Supplementary Table 5.

typically available across biobanks. However, a definition that maximizes data availability might not be one that best reflects the genetic etiology of a specific disease.

Third, as a common concern for all studies on disease progression, we examined the impact of index event bias resulting from conditioning on individuals with the disease. While this is not the main focus of this study, we found that index event bias alone does not fully explain the lack of concordance between genetic effects on susceptibility and progression observed in our study. Our empirical observations, in comparison to various simulations, indicate a relatively low heritability

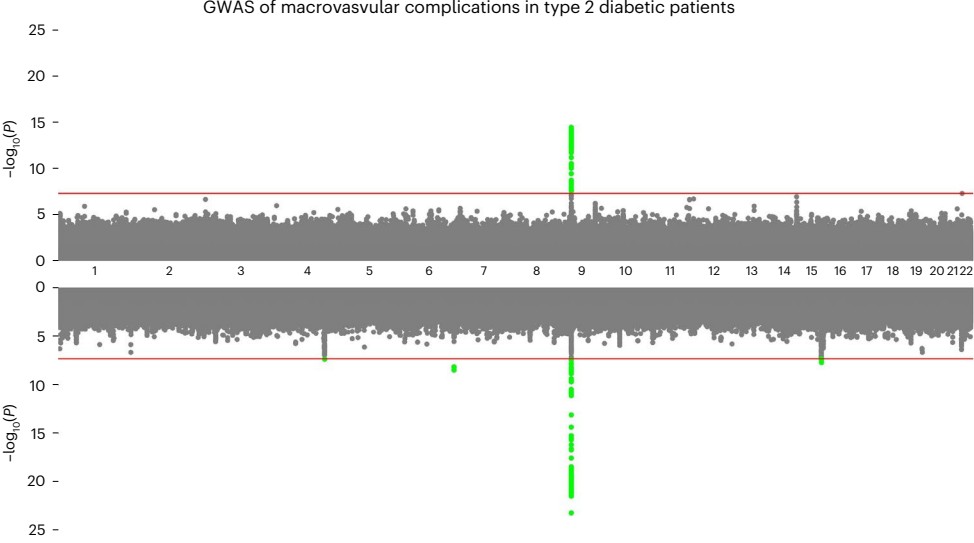

**Fig. 5 | GWAS of type 2 diabetes progression defined as macrovascular complications.** The top Manhattan plot displays the results of a GWAS carried out in individuals with type 2 diabetes. The bottom Manhattan plots display the results of GWAS carried out in individuals without type 2 diabetes, where proxy phenotypes for cardiovascular conditions were used instead. Two GWASs were matched to the same number of cases and controls to ensure similar power ($n_{cases}$ = 17,332 of 85,188 eligible type 2 diabetes patients/type 2 diabetes-free individuals).

for disease progression, as defined by disease-specific mortality. Furthermore, heterogeneous phenotypes, such as mortality, although disease-specific, can be highly polygenic. In this case, even a perfect correction for index event bias will only be able to recover effect sizes that are not likely to be detected from a progression GWAS. The fact that only one genome-wide significant locus was detected from our progression GWASs, indicating low signal-to-noise ratio in the progression GWASs, might be a bigger concern than index event bias. Furthermore, most methods to correct for index event bias rely on fitting the relationships between variant effects on susceptibility and progression. In our case, this relationship is close to zero; thus, the correction will be small and insignificant.

Lastly, even with all three aspects taken into account, there is still the possibility that we do not observe comparable genetic effects on disease susceptibility and progression simply because their underlying biological mechanisms are truly distinct.

Given the aforementioned challenges in conducting GWAS of disease progression, one practical alternative would be to study genetic signals for disease progression in a general population and subsequently adapt them for within-patient prognostic prediction. For example, PGSs for autoimmune conditions derived from the general population are correlated with immune-related adverse events among cancer patients treated with immune checkpoint inhibitors[39–42], and a PGS for ulcerative colitis was associated with immune checkpoint inhibitor-mediated colitis[42]. In our analysis, a longevity PGS derived from a GWAS of lifespan was substantially associated with survival for seven of nine diseases, suggesting that survival may be more affected by general factors related to mortality than by disease-specific factors. Similarly, the substantial findings for macrovascular complications in type 2 diabetes were recapitulated in the general population when looking at similar phenotypes, even among individuals without type 2 diabetes. In fact, the signal was stronger among individuals without type 2 diabetes, possibly reflecting index event bias. Notably, PGSs for cardiovascular conditions in the general population were better predictors of type 2 diabetes macrovascular complications than a PGS for type 2 diabetes susceptibility. Methods for cross-trait PGS[43] might be leveraged to obtain progression PGS based on existing GWAS results in the general population. This relationship, however, is not always

obvious. For diabetic microvascular complications, it is not as straightforward to find any population equivalent measurement, although it was interesting to observe a nominally substantial but weak effect of a PGS for age-related macular degeneration. On the other hand, this could indicate that such a progression definition is more unique to the diseased cohort and has the potential to yield truly progression-specific genetic signals, if sufficient power is available.

This study has multiple limitations. First, while we explored the similarity in genetic effects between disease susceptibility and disease-specific mortality, we cannot decisively conclude that the biological underpinnings of susceptibility and progression are distinct. For example, a phenotype that serves as poor proxy for disease progression will result in attenuated effect sizes, despite genetic variants being causally associated with both susceptibility and progression. Nonetheless, the poor replication rate and opposite direction of effect observed for susceptibility signals on disease-specific mortality are consistent with a scenario where at least some variants have no shared effect on both susceptibility and progression. Second, our findings do not necessarily extend outside the diseases explored in this study, and further work is needed to confirm the observed trends across more disease categories. Third, whether death certificates accurately capture primary or contributing causes of death depends on the biobank and healthcare system. We tried to address these concerns by restricting the follow-up duration in sensitivity analyses, reasoning that deaths occurring shortly after disease diagnosis were more likely to be caused by the disease.

In conclusion, our current results suggest that there is a limited overlap in genetic effects on disease susceptibility and progression, as defined by patients' mortality. Further refinement in inclusion criteria among the patient population and in the definitions of disease progression can be considered in future studies to robustly identify the genetic underpinnings of disease progression.

## Online content

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

Zhiyu Yang [1], Fanny-Dhelia Pajuste [2], Kristina Zguro[3], Yipeng Cheng[4], Danielle E. Kurant[5], Andrea Eoli [6,7], Julian Wanner [1,6], Bradley Jermy[1], Joel Rämö [1,5], FinnGen*, Stavroula Kanoni [8], David A. van Heel [9], Genes & Health Research Team*, Caroline Hayward [4], Riccardo E. Marioni [4], Daniel L. McCartney [4], Alessandra Renieri [3,10,11], Simone Furini[3,10,12], INTERVENE consortium*, Reedik Mägi[2], Alexander Gusev [5], Petros Drineas[13], Peristera Paschou[14], Henrike Heyne [1,6,7], Samuli Ripatti [1,15,16], Nina Mars [1,15] & Andrea Ganna [1,15,17] ✉

[1]Institute for Molecular Medicine Finland, FIMM, HiLIFE, University of Helsinki, Helsinki, Finland. [2]Estonian Genome Centre, Institute of Genomics, University of Tartu, Tartu, Estonia. [3]Med Biotech Hub and Competence Center, Department of Medical Biotechnologies, University of Siena, Siena, Italy. [4]Centre for Genomic and Experimental Medicine, Institute of Genetics and Cancer, University of Edinburgh, Edinburgh, UK. [5]Medical Oncology, Dana–Farber Cancer Institute, Harvard Medical School, Boston, MA, USA. [6]Hasso Plattner Institute, Digital Health Cluster, University of Potsdam, Potsdam, Germany. [7]Hasso Plattner Institute, Mount Sinai School of Medicine, New York City, NY, USA. [8]William Harvey Research Institute, Barts and the London School of Medicine and Dentistry, Queen Mary University of London, London, UK. [9]Blizard Institute, Barts and The London School of Medicine, Queen Mary University of London, London, UK. [10]Genetica Medica, Azienda Ospedaliera Universitaria Senese, Siena, Italy. [11]Medical Genetics, University of Siena, Siena, Italy. [12]Department of Electrical, Electronic and Information Engineering 'Guglielmo Marconi', University of Bologna, Cesena, Italy. [13]Department of Computer Science, Purdue University, West Lafayette, IN, USA. [14]Department of Biological Sciences, Purdue University, West Lafayette, IN, USA. v[15]Analytic and Translational Genetics Unit, Massachusetts General Hospital, Boston, MA, USA. [16]Department of Public Health, University of Helsinki, Helsinki, Finland. [17]Faculty of Medicine, University of Helsinki, Helsinki, Finland. *Lists of authors and their affiliations appear at the end of the paper. ✉e-mail: andrea.ganna@helsinki.fi

**FinnGen**

Zhiyu Yang[1], Henrike Heyne[1,6,7], Samuli Ripatti[1,15,16], Nina Mars[1,15], Andrea Ganna[1,15,17] & Estonian Biobank Research Team*

**Genes & Health Research Team**

Stavroula Kanoni[8] & David A. van Heel[9]

**INTERVENE consortium**

Zhiyu Yang[1], Fanny-Dhelia Pajuste[2], Kristina Zguro[3], Yipeng Cheng[4], Julian Wanner[1,6], Bradley Jermy[1], Stavroula Kanoni[8], Alessandra Renieri[3,10,11], Simone Furini[3,10], Reedik Mägi[2], Henrike Heyne[1,6,7], Samuli Ripatti[1,15,16], Nina Mars[1,15] & Andrea Ganna[1,15,17]

## Methods

### Ethics statement

This study was conducted in compliance with the relevant ethical guidelines and approved by the appropriate ethics committees. Details of the ethics committees of each participating biobank are provided in the Acknowledgements.

### Selection of diseases

We selected nine common complex diseases spanning various disease categories for the analyses. The diseases are selected to meet following criteria: (1) have high epidemiological HR on mortality, so that mortality can be viewed as a reasonable prognosis; (2) constitute high global disease burden in terms of disability adjusted life years[44]; (3) be relatively common ( > 1% prevalence) in population and have reasonable patient bodies in all biobanks and (4) be heritable and have large-scale GWAS available to construct PGSs. All disease endpoints were defined as a composition of ICD-10 codes curated by the clinical expert groups from FinnGen, Institute for Molecular Medicine Finland and Finnish Institute for Health and Welfare[19]. The same disease definitions, in terms of ICD-10 codes, were adopted by all participating biobanks to the maximum possible extent. See Supplementary Table 2 for a list of diseases and relevant descriptive statistics.

### Progression definition

For all selected diseases, we defined mortality as our outcome. Precisely, we were interested in both all-cause mortalities, namely simple death status of the patient regardless of relevance to the disease, and disease-specific mortalities, meaning the death caused directly or indirectly by disease of interest specifically. Disease progression was evaluated as patients' survival from each type of mortality after being diagnosed with the disease. For all mortality GWASs, we consider only disease-specific mortality whenever possible for each participating biobank, whereas for the PGS analysis, both all-cause and disease-specific mortalities were evaluated. Similar to the disease endpoints, cause of death linked to each disease was also curated by clinical expert groups and defined in terms of ICD-10 codes[45]. The same definitions were systematically applied to all biobanks to the possible extent. See Supplementary Table 2 for definitions of cause-specific mortality for each disease of interest and available sample sizes from each biobank.

### Within-patient mortality GWAS

To achieve variant-level effect comparison, a within-patient mortality GWAS was carried out for each selected disease using GATE26 for all biobanks, except Generation Scotland, which used SPACox[27] as an alternative. The event of interest in this GWAS was patients' survival after disease diagnosis. For each disease of interest, GWAS was carried out separately within each ancestry group for biobanks that have a cause-specific mortality event count of 50 at minimum after quality control. Eligible individuals were restricted to patients having a follow-up time after diagnosis of three months (0.25 years) at minimum. We used the model below to examine SNP association with patients' survival:

surv(duration of follow-up after diagnosis | disease-specific mortality) ~ SNP + patient's age of diagnosis + patient's birth year + sex + PCs + study-specific covariates,

where study-specific covariates included other available nonheritable biobank-specific covariates, such as genotyping chip or batch.

For analyses in the UK Biobank, to minimize potential impact of survivor bias, only patients with disease diagnosed after enrollment were considered.

### Results quality control and meta-analysis

After conducting mortality GWAS for selected diseases within each contributing biobank, we then filtered the resulting summary statistics by imputation INFO scores and minor allele counts. We retained only variants with an imputation INFO score >0.7 and at least 20 minor allele counts for each summary statistic. For GWAS summary statistics with a different human genome build, we used the UCSC LiftOver tool[46] to convert their genome coordinates into the hg38 assembly. Subsequently, for each disease, we meta-analyzed GWAS results from each biobank using fixed-effect meta-analysis implemented in METAL[47], with which we also scanned for heterogeneity in effect sizes across different biobanks using Cochran's $Q$ test. We applied an inverse-variance weighted meta-analysis scheme whenever possible. However, since SPACox does not have effect size or s.e. output, in Generation Scotland, we estimated direction of effect under a logistic regression model using PLINK[48] and subsequently proceeded with a sample-size weighted meta-analysis using the $Z$-scores. This was done for four of the nine diseases for which Generation Scotland was one of the data sources: atrial fibrillation, breast cancer, coronary artery disease and type 2 diabetes.

### Variant-level effect size comparison

We compared our mortality GWAS results for each disease of interest with large-scale published GWAS on diagnosis of the same disease. For disease diagnosis GWAS, we extracted SNP effects of reported genome-wide significant leading SNPs at independently associated loci from each study. For chronic kidney disease, a large GWAS on estimated glomerular filtration rate was considered[34]. Specifically, we examined the effect sizes of independent lead SNPs on the binary diagnosis of chronic kidney disease reported in the study, ensuring a more comparable scale of measurement. For our meta-analyzed mortality GWAS, we identify independent genome-wide loci using summary statistics based on conditional analysis implemented in GCTA-COJO. We merged 5,000 Finnish genomes, which is one of the largest GWAS cohorts in this study, with EUR from Human Genome Diversity Project as linkage disequilibrium (LD) reference for this step. To carry out the effect size comparison for all diseases, we reran the meta-analysis of mortality GWAS, excluding results from Generation Scotland due to the use of an incomparable GWAS approach for the cohort.

### Comparison of genetic architectures

We compared genetic architectures between disease diagnosis and mortality in terms of SNP heritability estimated from the meta-analyzed mortality GWAS summary statistics using LD score regression[49]. For eligible traits, that is, traits with nonzero estimated SNP heritability, we further analyzed genetic correlation across disease diagnosis, mortality, and general longevity GWAS using the same tool.

### Down-sampled GWAS on age of diagnosis

To ensure heritability comparison between disease susceptibility and progression endpoints not being subject to power issues resulting from difference in sample sizes and GWAS models, for each disease of interest, we also ran time-to-event GWAS to find SNP association with age of diagnosis using a randomly down-sampled cohort which had comparable number of total individuals and event counts as what was available for the within-patient mortality GWAS. The down-sampled GWAS was carried out under the model below:

surv(follow-up from birth until diagnosis | disease diagnosis) ~ SNP + patient's birth year + sex + PCs + study-specific covariates.

This analysis was also carried out using GATE[26] but in FinnGen and UK Biobank only, which are two of the largest participating biobanks in this study (see Supplementary Table 2 for sample sizes).

### Computation of individual-level PGS

For each selected disease, we derived variant weights for PGSs from GWAS summary statistics listed in Supplementary Table 2 using MegaPRS[50]. Heritability contributed by each variant was estimated under the BLD-LDAK model as recommended. For weight estimation, we used the 'mega' option, which allows the software to determine the

most appropriate model based on the data. Since we studied mortality, apart from the nine selected diseases, we also computed PGS weights for general longevity using the largest GWAS on lifespan[29]. Due to the heterogeneous and polygenic nature of lifespan, we used the LDAK-Thin model for SNP-level heritability estimation for this trait instead. Unlike the BLD-LDAK model used in variant weighting for other diseases, LDAK-Thin model does not take functional annotations into account but estimates SNP heritability only as function of SNP allele frequencies and local linkage structures. Variant weights were derived for 1,330,820 common SNPs (minor allele frequency > 0.1) lying in the intersection of HapMap3 (ref. [51]) and 1000 Genomes[52] that are available for each GWAS summary statistic.

Once the SNP weights were derived, individual-level PGSs for each disease and general longevity were subsequently computed as a weighted sum of effect allele counts using PLINK[48]. Scores were standardized to have 0 mean and 1 as variance within each ancestry group.

For the composite mortality PGS, we used sex-stratified SNP weights developed by ref. [30]. Scores for males and females were computed separately and subsequently combined during the association step to obtain a population effect estimate.

## Association between PGS and disease of interest

As a baseline, we first examined whether the disease PGSs were associated with their diagnoses. For each selected disease, the association was first tested using a general linear model on case–control status as below:

logit(Pr(Individual is diagnosed)) ~ disease PGS + birth year + sex + PC1-10.

To achieve a fairer comparison with the other experiments, we also evaluated such relationship using a survival model on the age of diagnosis as below:

surv(follow-up from birth until diagnosis | disease diagnosis) ~ disease PGS + birth year + sex + PC1-10.

The two analyses above were conducted using all eligible individuals from the biobanks. Then, for each selected disease, we extracted only the patient group for further analysis. To reduce noise in measurements, we limited these within-patient analyses to individuals having a follow-up time of at least three months (0.25 years) after the diagnosis. We tested the association of disease PGSs with our defined prognosis, namely patient survival, using the model below:

surv(duration of follow-up after diagnosis | mortality) ~ disease PGS + birth year + sex + PC1-10 + age of diagnosis,

as well as the association of general longevity PGS with patient survival as below:

surv(duration of follow-up after diagnosis | mortality) ~ general longevity PGS + birth year + sex + PC1-10 + age of diagnosis.

For both associations, we examined both all-cause mortality and cause-specific mortality within the patient group. All analyses were corrected for sex, except in analyses for breast cancer and prostate cancer, where only female/male individuals were used.

These analyses were carried out independently for each ancestry group within each participating biobank. We only included biobanks where the count of events of interest in the analyzed ancestry group was 50 or more. We subsequently meta-analyzed effect sizes for the same ancestry group across biobanks using the inverse-variance weighted approach.

## Mortality PGSs and their performance in FinnGen

For diseases with sufficient power, we derived mortality PGS weights using meta-analyzed mortality GWAS results of European populations from all available biobanks, except for FinnGen or Generation Scotland. Apart from FinnGen, which was used as a test cohort, we also left out results from Generation Scotland for this analysis because their summary statistics did not have effect size or s.e. and therefore cannot be used for inverse-variance weighted

meta-analysis, which returns necessary statistics for weight derivation. After deriving PGS weights using MegaPRS[50], we subsequently computed individual-level disease-mortality PGS for patients of each corresponding disease within FinnGen cohort. The weights and scores are computed in the same manner as mentioned in the 'Computation of individual-level PGS'. We evaluated the effects of these scores on predicting patients' disease mortality in FinnGen using the model below:

surv(duration of follow-up after diagnosis | mortality) ~ disease-mortality PGS + birth year + sex + PC1-10 + age of diagnosis

## Sensitivity analyses for PGS experiments

We ran a series of sensitivity analyses in eligible biobanks to ensure our observations on the PGSs association were robust, under considerations listed below. Similarly, analyses were conducted for each eligible ancestry within each biobank and then meta-analyzed.

First, to demonstrate the impact of relevance between disease progression and susceptibility as shown in our theories, we examined the association between susceptibility PGS and all-cause mortality and compared the results with disease-specific mortality in FinnGen (see Supplementary Fig. 10 for these results). We then considered other factors that may bias the results.

**Survivor bias.** Depending on each biobank's recruitment scheme, some patients were diagnosed before the start of their follow-up, which may lead to biased results due to the survivor effect. Therefore, we also ran these analyses for each disease using only samples from individuals enrolled before their first onset of the disease of interest (see Supplementary Fig. 11a for these results).

**Relevance between cause of mortality in death certificate and disease diagnosis.** In this study, we aimed to define disease progression as accurately as possible by focusing our analysis on disease-caused mortality. However, some national death registries may not precisely capture the immediate cause of death, and some mortalities, while documented with the disease as one of the causes, may not be truly relevant to the diagnosed disease. To address this concern, we ran the same analysis using only patients with a restricted maximum follow-up length, since death taking place reasonably soon after being diagnosed might have more to do with the diagnosis, compared to death taking place decades after. Under this consideration, we varied the maximum duration of follow-up after diagnosis by 2, 5 or 10 years. The minimum is still 0.25 years for this analysis (see Supplementary Fig. 11b and Supplementary Table 8 for these results; see also Supplementary Table 2 for sample size breakdown by duration of follow-up in each biobank). To facilitate comparability between results, we reported the regression coefficients for PGS effect sizes on nine diseases for each sensitivity analysis and the main results.

**The effect of diagnosed age.** As shown above, we included the age of diagnosis as one of the covariates in all within-patient main analysis models to specifically investigate PGSs' unique genetic effect on disease progression by correcting for the diagnosis. As part of our sensitivity analysis, we also examined the role of these diagnosed ages in more detail. We repeated all the within-patient analyses for each disease by stratifying patients into early onset and late onset groups using 50% age of diagnosis quantile as a cutoff and compared the PGS effects across the two groups (see Supplementary Fig. 12 and Supplementary Table 9 for these results).

**Sample relatedness.** We included all eligible individuals of each biobank in our main analysis, and one may argue that this could impact our effect size estimates. Therefore, we ran the same analysis in FinnGen with up to second-degree relatives removed (see Supplementary Fig. 13 and Supplementary Table 7 for these results).

**Results from non-European ancestry populations.** Since only patients were considered for most of our analyses, although some of the biobanks (for example, UK Biobank and BioMe) were known to be rather diverse, we ended up with enough power for the main results only for the European super-population. Nevertheless, comparison of results with other less powered but available populations can be found in Supplementary Fig. 15 for reference.

Forest plot for effects from each biobank is presented in Supplementary Fig. 14.

### Alternative progression definitions for type 2 diabetes

For type 2 diabetes, we explored the genetics of two additional widely considered progressions−macrovascular and microvascular complications. For macrovascular complications, we only consider patients who did not have any coronary artery disease, stroke or peripheral arterial disease incidents before the onset of type 2 diabetes. Among those, we define the ones having at least one of the aforementioned diagnoses after type 2 diabetes as cases for macrovascular complications. Event time is defined as the duration from a patient's diagnosis of type 2 diabetes to the earliest diagnosis of a macrovascular complication. Similarly, for microvascular complications, we consider onset of diabetic retinopathy, nephropathy and neuropathy after the patients' diagnosis of type 2 diabetes. For both definitions of progression, our analysis only included individuals with >0.25 year of follow-up, meaning the patients' death/onset of progression/biobank censoring take place >0.25 year after their diagnosis of type 2 diabetes.

For macrovascular complications, for which we identified genome-wide significant signals among diabetic patients, we further carried out a down-sampled time-to-event GWAS on population-comparable phenotypes, matching the case–control count in the progression GWAS. For this down-sampled GWAS, we considered onset of coronary artery disease, stroke, or peripheral arterial disease in nondiabetic population.

### Simulation to explore the impact of index event bias

Please see section 'Simulation to explore the impact of index event bias' from Supplementary Note for details.

### Reporting summary

Further information on research design is available in the Nature Portfolio Reporting Summary linked to this article.

## Data availability

Summary statistics of meta-analyzed disease-mortality GWASs for nine diseases of interest can be found from figshare public project https://figshare.com/projects/Progression_GWAS/252002 (ref. 53). Weights used to compute susceptibility and longevity PGSs can be downloaded from https://github.com/Zhiyu-9668/ProgressAnalysis/.

## Code availability

Code used for within-biobank PGS association analyses, results meta-analysis and plotting can be found on public GitHub repo https://github.com/Zhiyu-9668/ProgressAnalysis/ and Zenodo under https://doi.org/10.5281/zenodo.16946874 (ref. 54).

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

## Acknowledgements

This study received funding from the European Union's Horizon 2020 research and innovation program (grant agreement 101016775 to S.R. and A.G.), from the European Research Council (ERC) under the European Union's Horizon 2020 research and innovation program (grant 945733 to A.G.) and from Academy of Finland fellowship (grant 323116 to A.G.). We thank George Davey Smith for the invaluable comments on this manuscript. We thank all those who contributed samples and data for the FinnGen scientific project and P. VandeHaar for technical consultation on PheWeb. The FinnGen project is funded by two grants from Business Finland (HUS 4685/31/2016 and UH 4386/31/2016) and the following industry partners: AbbVie, AstraZeneca UK, Biogen, Bristol Myers Squibb (and Celgene Corporation & Celgene International II), Genentech, Merck Sharp & Dohme, a subsidiary of Merck & Co., Pfizer, GlaxoSmithKline Intellectual Property Development, Sanofi US Services, Maze Therapeutics, Janssen Biotech, Novartis and Boehringer Ingelheim. The following biobanks are acknowledged for delivering samples to FinnGen: Auria Biobank (https://www.auria.fi/biopankki/), THL Biobank (https://www.thl.fi/biobank), Helsinki Biobank (https://www.helsinginbiopankki.fi), Biobank Borealis of Northern Finland (https://oys.fi/biopankki/briefly-in-english/), Finnish Clinical Biobank Tampere (https://www.tays.fi/en-US/Research_and_development/Finnish_Clinical_Biobank_Tampere), Biobank of Eastern Finland (https://www.ita-suomenbiopankki.fi/en), Central Finland Biobank (https://www.ksshp.fi/fi-FI/Potilaalle/Biopankki), Finnish Red Cross Blood Service Biobank (www.veripalvelu.fi/verenluovutus/biopankkitoiminta) and Terveystalo Biobank (https://www.terveystalo.com/fi/Yritystietoa/Terveystalo-Biopankki/Biopankki/). All Finnish biobanks are members of the BBMRI.fi infrastructure (https://www.bbmri.fi). The FINBB (https://finbb.fi/) is the coordinator of BBMRI-ERIC operations in Finland. The Finnish biobank data can be accessed through the Fingenious services (https://site.fingenious.fi/en/) managed by FINBB. Ethics approval for the UK Biobank study was obtained from the North West Centre for Research Ethics Committee (11/NW/0382). UK Biobank data used in this study were obtained under application 78537. We also acknowledge all the recruiters and the participants of the EGCUT, the University of Tartu, the Ministry of Social Affairs, the Ministry of Science and Education, the Ministry of Economic Affairs and Communications, the Archimedes Foundation, the Estonian Biocentre, the Institute of Molecular and Cell Biology and the Centre for Ethics of the University of Tartu. We also acknowledge the EGCUT

technical personnel. Other contributors were A. Allik, T. Annilo, M. Hass, A.-H. Jõks, A.-T. Kaasik, A. Keis, E. Leego, M. Leego, K. Lilienthal, K. Metsalu, E. Mihailov, K. Mikkel, E. Mölder, H. Niinemäe, T. Nikopensius, M. Puusepp, S. Smit, V. Soo, R. Tamm, M. Teder-Laving and M. Väli-Täht. This research was made possible through access to data in the National Genomic Research Library, which is managed by Genomics England Limited (a wholly owned company of the Department of Health and Social Care). The National Genomic Research Library holds data provided by patients and collected by the National Health Service (NHS) as part of their care and data collected as part of their participation in research. The National Genomic Research Library is funded by the National Institute for Health Research and NHS England. The Wellcome Trust, Cancer Research UK and the Medical Research Council have also funded research infrastructure. We acknowledge the contribution of the Genomics England Research Consortium. The members of this consortium are J. C. Ambrose, P. Arumugam, R. Bevers, M. Bleda, F. Boardman-Pretty, C. R. Boustred, H. Brittain, M. J. Brown, M. J. Caulfield, G. C. Chan, A. Giess, A. Hamblin, S. Henderson, T. J. P. Hubbard, R. Jackson, L. J. Jones, D. Kasperaviciute, M. Kayikci, A. Kousathanas, L. Lahnstein, S. E. A. Leigh, I. U. S. Leong, J. F. Lopez, F. Maleady-Crowe, M. McEntagart, F. Minneci, J. Mitchell, L. Moutsianas, M. Mueller, N. Murugaesu, A. C. Need, P. O'Donovan, C. A. Odhams, C. Patch, D. Perez-Gil, M. B. Pereira, J. Pullinger, T. Rahim, A. Rendon, T. Rogers, K. Savage, K. Sawant, R. H. Scott, A. Siddiq, A. Sieghart, S. C. Smith, A. Sosinsky, A. Stuckey, M. Tanguy, A. L. T. Tavares, E. R. A. Thomas, S. R. Thompson, A. Tucci, M. J. Welland, E. Williams, K. Witkowska, S. M. Wood and M. Zarowiecki. Generation Scotland received core support from the Chief Scientist Office of the Scottish Government Health Directorates (CZD/16/6) and the Scottish Funding Council (HR03006). Genotyping of the GS:SFHS samples was carried out by the Genetics Core Laboratory at the Wellcome Trust Clinical Research Facility, Edinburgh, Scotland, and was funded by the Medical Research Council, UK, and the Wellcome Trust (Wellcome Trust Strategic Award 'Stratifying Resilience and Depression Longitudinally' (STRADL) reference 104036/Z/14/Z). We are grateful to all the families who took part, the general practitioners and the Scottish School of Primary Care for their help in recruiting them and the whole Generation Scotland team, which includes interviewers, computer and laboratory technicians, clerical workers, research scientists, volunteers, managers, receptionists, healthcare assistants and nurses. Ethical approval for the GS:SFHS study was obtained from the Tayside Committee on Medical Research Ethics (on behalf of the National Health Service). Ethical approval for the GS:3D study was obtained from the Tayside Committee on Medical Research Ethics (on behalf of the National Health Service). Ethical approval for the GS:21CGH study was obtained from the Scotland A Research Ethics Committee. Genes & Health is/has recently been core-funded by Wellcome (WT102627 and WT210561), the Medical Research Council (UK; M009017, MR/X009777/1 and MR/X009920/1), Higher Education Funding Council for England Catalyst, Barts Charity (845/1796), Health Data Research UK (for London substantive site) and research delivery support from the NHS National Institute for Health Research Clinical Research Network (North Thames). Genes & Health is/has recently been funded by Alnylam Pharmaceuticals, Genomics, and a Life Sciences Industry Consortium of AstraZeneca, Bristol Myers Squibb Company, GlaxoSmithKline Research and Development, Maze Therapeutics, Merck Sharp & Dohme, Novo Nordisk A/S, Pfizer and Takeda Development Center Americas. We thank Social Action for Health, Centre of The Cell, members of our Community Advisory Group and staff who have recruited and collected data from volunteers. We thank the NIHR National Biosample Centre (UK Biocentre); the Social, Genetic & Developmental Psychiatry Centre (King's College London); Wellcome Sanger Institute; and Broad Institute for sample processing, genotyping, sequencing and variant annotation. We thank Barts Health NHS Trust, NHS Clinical Commissioning Groups (City and Hackney, Waltham Forest, Tower Hamlets, Newham, Redbridge, Havering, Barking and Dagenham), East London NHS Foundation Trust, Bradford Teaching Hospitals NHS Foundation Trust, Public Health England (especially David Wyllie), Discovery Data Service/Endeavour Health Charitable Trust (especially David Stables), Voror Health Technologies (especially Sophie Don) and NHS England (for what was NHS Digital)—for General Data Protection Regulation-compliant data sharing backed by individual written informed consent. This work was supported in part by the computational and data resources, and staff expertise provided by Scientific Computing and Data at the Icahn School of Medicine at Mount Sinai and supported by the Clinical and Translational Science Awards (CTSA; grant UL1TR004419) from the National Center for Advancing Translational Sciences. Additionally, this work was supported by the Office of Research Infrastructure of the National Institutes of Health (award S10OD026880), which allowed us to use Mount Sinai Data Warehouse (MSDW) data. Regarding HPI.MS resources, funding was provided by the Hasso Plattner Foundation (HPF). The Mount Sinai BioMe Biobank has been supported by The Andrea and Charles Bronfman Philanthropies and, in part, by Federal funds from the The National Heart, Lung, and Blood Institute and National Human Genome Research Institute (U01HG00638001, U01HG007417 and X01HL134588). We thank all participants in the Mount Sinai BioMe Biobank. We also thank all of our recruiters who have assisted in data collection and management and are grateful for the computational resources and staff expertise provided by Scientific Computing at the Icahn School of Medicine at Mount Sinai.

## Author contributions

Z.Y. designed and led the analyses, analyzed FinnGen and UK Biobank data, meta-analyzed results and ran downstream analyses, and wrote the manuscript. F.-D.P., K.Z., Y.C., D.E.K., A.E. and J.W. were biobank lead analysts who ran the GWAS, ranked by biobank sample sizes. B.J. provided weights for susceptibility PGSs. J.R. provided valuable clinical input for type 2 diabetes analyses. S.K. and D.A.v.H. supervised and led GWAS analysis in Genes & Health. C.H., R.E.M., D.L.M., A.R., S.F. and R.M. supervised analyses in each represented biobank. A.G., P.D. and P.P. provided valuable input on the theoretical framework and clinical impact. H.H., S.R., N.M. and A.G. designed and supervised the study. All authors contributed to revising the manuscript draft before submission.

## Funding

## Competing interests

A.G. is the founder of Real World Genetics Oy. The other authors declare no competing interests.

## Additional information

**Correspondence and requests for materials** should be addressed to Andrea Ganna.

# Reporting Summary

## Statistics

For all statistical analyses, confirm that the following items are present in the figure legend, table legend, main text, or Methods section.

| n/a | Confirmed | |
|---|---|---|
| ☐ | ☒ | The exact sample size (*n*) for each experimental group/condition, given as a discrete number and unit of measurement |
| ☐ | ☒ | A statement on whether measurements were taken from distinct samples or whether the same sample was measured repeatedly |
| ☐ | ☒ | The statistical test(s) used AND whether they are one- or two-sided<br>*Only common tests should be described solely by name; describe more complex techniques in the Methods section.* |
| ☐ | ☒ | A description of all covariates tested |
| ☐ | ☒ | A description of any assumptions or corrections, such as tests of normality and adjustment for multiple comparisons |
| ☐ | ☒ | A full description of the statistical parameters including central tendency (e.g. means) or other basic estimates (e.g. regression coefficient) AND variation (e.g. standard deviation) or associated estimates of uncertainty (e.g. confidence intervals) |
| ☐ | ☒ | For null hypothesis testing, the test statistic (e.g. *F*, *t*, *r*) with confidence intervals, effect sizes, degrees of freedom and *P* value noted<br>*Give P values as exact values whenever suitable.* |
| ☒ | ☐ | For Bayesian analysis, information on the choice of priors and Markov chain Monte Carlo settings |
| ☐ | ☒ | For hierarchical and complex designs, identification of the appropriate level for tests and full reporting of outcomes |
| ☐ | ☒ | Estimates of effect sizes (e.g. Cohen's *d*, Pearson's *r*), indicating how they were calculated |

*Our web collection on statistics for biologists contains articles on many of the points above.*

## Software and code

Policy information about availability of computer code

| Data collection | See supplementary notes for biobank specific data collection approach. |
|---|---|
| Data analysis | Major tools used in data analysis include: PLINK v2, GATE v0.45, SPACox v2.1.0, MegaPRS v5.1, LDSC (LD SCore) v1.0.1, and METAL (version 2011-03-25). |

For manuscripts utilizing custom algorithms or software that are central to the research but not yet described in published literature, software must be made available to editors and reviewers. We strongly encourage code deposition in a community repository (e.g. GitHub). See the Nature Portfolio guidelines for submitting code & software for further information.

## Data

Policy information about availability of data

All manuscripts must include a data availability statement. This statement should provide the following information, where applicable:

- Accession codes, unique identifiers, or web links for publicly available datasets
- A description of any restrictions on data availability
- For clinical datasets or third party data, please ensure that the statement adheres to our policy

Summary statistics for mortality GWAS of phenotypes of interest can be found in figshare public project https://figshare.com/projects/Progression_GWAS/252002
PGS weights and analyses code can be found in public github repo https://github.com/Zhiyu-9668/ProgressAnalysis

# Research involving human participants, their data, or biological material

Policy information about studies with [human participants or human data](). See also policy information about [sex, gender (identity/presentation), and sexual orientation]() and [race, ethnicity and racism]().

| | |
|---|---|
| Reporting on sex and gender | Biological sex has been used as a covariate in GWAS and PGS related analysis. Gender is not of interest in this study. |
| Reporting on race, ethnicity, or other socially relevant groupings | Race group in the study was determined using PCA or comparable method. Analysis was stratified by grouped race due to cross-group heterogeneity. |
| Population characteristics | Cohort specific disease and mortality information can be found in supplementary table 2. |
| Recruitment | Recruitment strategy varies across participating biobanks. We only consider disease onset after enrollment whenever possible to take into account of possible survivor bias. |
| Ethics oversight | The study was conducted in compliance with the relevant ethical guidelines and approved by the appropriate ethics committees. Ethics approval for the UK Biobank study was obtained from the North West Centre for Research Ethics Committee (11/NW/0382). Ethical approval for the GS:SFHS study was obtained from the Tayside Committee on Medical Research Ethics (on behalf of the National Health Service). Ethical approval for the GS:3D study was obtained from the Tayside Committee on Medical Research Ethics (on behalf of the National Health Service). Ethical approval for the GS:21CGH study was obtained from the Scotland A Research Ethics Committee. Ethical approval for the GS:SFHS study was obtained from the Tayside Committee on Medical Research Ethics (on behalf of the National Health Service). Ethical approval for the GS:3D study was obtained from the Tayside Committee on Medical Research Ethics (on behalf of the National Health Service). Ethical approval for the GS:21CGH study was obtained from the Scotland A Research Ethics Committee.Ethical approval for the GS:SFHS study was obtained from the Tayside Committee on Medical Research Ethics (on behalf of the National Health Service). Ethical approval for the GS:3D study was obtained from the Tayside Committee on Medical Research Ethics (on behalf of the National Health Service). Ethical approval for the GS:21CGH study was obtained from the Scotland A Research Ethics Committee. |

Note that full information on the approval of the study protocol must also be provided in the manuscript.

# Field-specific reporting

Please select the one below that is the best fit for your research. If you are not sure, read the appropriate sections before making your selection.

☒ Life sciences ☐ Behavioural & social sciences ☐ Ecological, evolutionary & environmental sciences

For a reference copy of the document with all sections, see [nature.com/documents/nr-reporting-summary-flat.pdf]()

# Life sciences study design

All studies must disclose on these points even when the disclosure is negative.

| | |
|---|---|
| Sample size | We use the maximum available samples after quality control across all partnership. |
| Data exclusions | We consider only patients having a follow-up duration >= 3mo. after being diagnosed with each disease of interest to grant enough data coverage for included individuals. Depending on biobank recruitment strategy, we also consider only individuals diagnosed after enrollment whenever possible to take into account of possible survivor bias. |
| Replication | Not applicable since no robust signal from mortality GWAS to be replicated. |
| Randomization | Phenotypes from each biobank are defined using ICD code as listed in supplementary table 2. Randomization does not apply to this study design. |
| Blinding | Our study was not a controlled trial but a genome-wide association study (GWAS). Blinding and randomization do not apply. |

# Reporting for specific materials, systems and methods

We require information from authors about some types of materials, experimental systems and methods used in many studies. Here, indicate whether each material, system or method listed is relevant to your study. If you are not sure if a list item applies to your research, read the appropriate section before selecting a response.

## Materials & experimental systems

| n/a | Involved in the study |
|-----|----------------------|
| ☒ ☐ | Antibodies |
| ☒ ☐ | Eukaryotic cell lines |
| ☒ ☐ | Palaeontology and archaeology |
| ☒ ☐ | Animals and other organisms |
| ☒ ☐ | Clinical data |
| ☒ ☐ | Dual use research of concern |
| ☒ ☐ | Plants |

## Methods

| n/a | Involved in the study |
|-----|----------------------|
| ☒ ☐ | ChIP-seq |
| ☒ ☐ | Flow cytometry |
| ☒ ☐ | MRI-based neuroimaging |

## Plants

| Seed stocks | NA |
|-------------|----|

| Novel plant genotypes | NA |
|-----------------------|----|

| Authentication | NA |
|----------------|----|

