## [Peer Review File · Nature Genetics]

Limited overlap between genetic effects on disease susceptibility and disease survival

Corresponding Author: Professor Andrea Ganna

Version 0:

Decision Letter:

8th March 2024

Dear Andrea,

Your Article "Limited overlap between genetic effects on disease susceptibility and disease survival" has been seen by three referees. You will see from their comments below that, while they find your work of potential interest, they have raised substantial concerns that must be addressed. In light of these comments, we cannot accept the manuscript for publication at this time, but we would be interested in considering a substantially revised version that addresses their key concerns.

We hope you will find the referees' comments useful as you decide how to proceed. If you wish to submit a substantially revised manuscript, please bear in mind that we will be reluctant to approach the referees again in the absence of major revisions.

To guide the scope of the revisions, the editors discuss the referee reports in detail within the team, including with the chief editor, with a view to identifying key priorities that should be addressed in revision, and sometimes overruling referee requests that are deemed beyond the scope of the current study. In this case, we ask that you restructure the paper to focus on the empirical results, while de-emphasizing or removing the theoretical modeling, as recommended by Reviewers #1 and #2. We also ask that you address all technical points with appropriate revisions and revise the presentation for clarity where needed. We hope that you will find this prioritized set of referee points to be useful when revising your study. Please do not hesitate to get in touch if you would like to discuss these issues further.

If you choose to revise your manuscript taking into account all reviewer and editor comments, please highlight all changes in the manuscript text file. At this stage we will need you to upload a copy of the manuscript in MS Word .docx or similar editable format.

*2) If you have not done so already, please begin to revise your manuscript so that it conforms to our Article format instructions, available [here](http://www.nature.com/ng/authors/article_types/index.html). Refer also to any guidelines provided in this letter.

*3) Include a revised version of any required Reporting Summary: <https://www.nature.com/documents/nr-reporting-summary.pdf>
It will be available to referees (and, potentially, statisticians) to aid in their evaluation if the manuscript goes back for peer review.

Link Redacted

If you wish to submit a suitably revised manuscript, we hope to receive it within 3-6 months. If you cannot send it within this time, please let us know. We will be happy to consider your revision so long as nothing similar has been accepted for publication at Nature Genetics or published elsewhere. Should your manuscript be substantially delayed without notifying us in advance and your article is eventually published, the received date would be that of the revised, not the original, version.

Nature Genetics is committed to improving transparency in authorship. As part of our efforts in this direction, we are now requesting that all authors identified as 'corresponding author' on published papers create and link their Open Researcher and Contributor Identifier (ORCID) with their account on the Manuscript Tracking System (MTS), prior to acceptance. ORCID helps the scientific community achieve unambiguous attribution of all scholarly contributions. You can create and link your ORCID from the home page of the MTS by clicking on 'Modify my Springer Nature account'. For more information, please visit www.springernature.com/orcid.

Thank you for the opportunity to review your work.

Sincerely,
Kyle

Kyle Vogan, PhD
Senior Editor
Nature Genetics
<https://orcid.org/0000-0001-9565-9665>

Referee expertise:

Referee #1: Genetics, common diseases, statistical methods

Referee #2: Genetic epidemiology, cancer, statistical methods

Referee #3: Genetics, cardiovascular diseases, clinical translation

Reviewers' Comments:

Reviewer #1:
Remarks to the Author:

The article presents results from GWAS of disease-specific mortality for common complex diseases. This is the largest study of its kind, and the main message is that there is very little signal of genetic effects on disease progression among diagnosed patients. This is mostly consistent with other disease-specific efforts. This mostly null study points to challenges associated with finding genetic determinants of disease progression using the GWAS approach. The authors spend considerable time developing an abstract (unobservable) model that considers disease progression in the general population. While the math is correct and interesting, I was not convinced of the practical value of such thinking. Below are some specific comments.

The authors should provide estimates of heritability for disease-specific mortality and susceptibility across the different diseases side by side in a main table. Such a table will help readers understand the impact of genetic contribution on these two traits. It would be further useful if, in the table legend, the authors could remark on the relationship of these definitions of heritability, which are directly estimable, to the concepts of heritability defined in their mathematical framework. Currently, the results for disease-specific mortality are apparently presented in the Supplemental Table (S2), but I could not dig out those results because of the huge size of S1.

The authors seem to have missed a key reference to a prior study on polygenic modeling of all-cause mortality (longevity) using combinations of susceptibility genetic scores (Meisner et al., AJHG, 2020, PMID: 32758451). In fact, this defined a more powerful PGS for predicting longevity than the PGS that has been defined directly based on GWAS of longevity phenotype (Timmers et al., 2019). Meisner et al also reported the association of the susceptibility PRS with disease-specific mortality across the different diseases. All of these are very relevant for this paper.

Line 132-136. I am surprised by the hypothesis here that one expects the effects of SNPs to be the same on the disease risk vs disease progression. By default, I would expect they will not be the same. The more relevant hypothesis is that whether the susceptibility alleles have some kind of effect on progression and in what direction. I did not find Figure 2, which investigates the hypothesis of equal effect, to be interesting, and it should be moved to supplement.

The authors spend considerable effort developing an abstract mathematical model for disease progression in the general population, including individuals not diagnosed with the disease. The point being that ideally, one would like to estimate the effects of genetic variants on the disease progression in that abstract setting. I was not very convinced with this argument. The authors did not explain or demonstrate what is the practical utility of such models for obtaining insight into biology or, risk prediction, or both. Second, such a model cannot be fitted based on the observable data, so everything stays very hypothetical. Even in their own discussion (lines 319-323) regarding the cause for low heritability for disease progression, the authors argued for potential effects of heterogeneous treatments that are only relevant for the population of affected individuals. I do think the math is correct and interesting, but I failed to see a lot of practical value in it.

The authors mostly used simulation studies to examine the relationship between "idealized" and "observable" genetic effects on disease progression. I wondered if they could have used the mathematical results from the real datasets to say more about the parameters of the idealized model for disease progression in the general population. The only direct relevance of all the math in the interpretation of the results from real studies I could see is the investigation of index-event bias.

The supplementary methods section is unnecessarily detailed in places and can be shortened substantially. For example, the theoretical results must only be derived once assuming a general value of ρ , and then the formula for $\rho=0$ case can be derived as a special case. What is the point of introducing the problem through a general function $f(S)$ where all of the calculations actually require $f(S)$ to be linear? Also, there was a mention of β being sparse, but that is never needed in the calculations.

Reviewer #2:

Remarks to the Author:

This paper explores the genetic contribution to disease-specific survival across ten common diseases using a relatively large number of cases from seven biobanks. It finds little evidence for strong genetic association with disease-specific survival, suggesting that the strength of association between common variants and disease-specific survival is much weaker than that for common variants and disease incidence. (The paper includes a nice empirical demonstration that, in equivalent sample sizes, GWAS of disease incidence find many more associations than GWAS of disease progression.) The paper also looks at the association between polygenic scores for disease incidence and disease-specific mortality, finding at most modest associations. Finally, the paper proposes a mathematical model for the co-contribution of genetics to incidence and survival and runs simulations based on that model to draw inference about study designs best suited for the study of disease-specific survival, the likelihood and magnitude of index-event bias, etc.

Strengths: Although the individual disease sample sizes are not always as large as previously published studies of disease-specific survival (c.f. breast cancer), they are each relatively large and large in aggregate. Including ten diseases in one paper is nice, as it allows general conclusions and contrasts to be made. For example, it is interesting to see the direction of the slopes of the regression of survival effects on incidence effects vary from disease to disease in Figure 2 and to see the incidence PGS-survival associations vary across disease. (One who only knows the breast and prostate cancer genetic epidemiology literature might be tempted to rashly conclude there is a general inverse relationship between susceptibility and survival--Figures 2 and 3 will set them aright.)

Weaknesses: The "theoretical framework" sections are unclear, speculative, and, I suspect, contain mathematical errors.

Clarity: many readers will not catch the fact that the additive liability model introduced in the theoretical framework section does not have a clear relationship to the empirical results presented from proportional hazards regressions. (The "heritability" of a censored survival trait is rather tricky to define; there's not a natural way to partition variance into "genetic" and "environmental" components under a proportional hazards model.) The paper flags this difference near the end of the discussion, 100 lines later. Nor does the paper ever, anywhere that I have noticed, define what is meant by liability. One presumes based on tradition that there is a probit link relating probabilities of disease incidence and survival to the additive liability. Would be nice to see that explicitly stated.

Mathematical concerns: The theoretical framework section needs another close read for mathematical rigor. I had several concerns. First, the liability of within-patient disease progression is not $P|S$ or $P|S=s$, it is $P|S>k$, where k is the liability threshold for disease incidence. $P|S>k$ seems to be the latent phenotype of interest here, not $P|S=s$. Second, the notation $\text{Var}(P|S)$, $G_{P|S}$ etc. seems rather loose, and may lead some calculations astray. In standard prob/stats notation, $\text{Var}(P|S)$ is a random variable; specifically, it is a function of the random variable S . It is not a fixed quantity. In the $\rho=0$ case, the S drops out of $\text{Var}(P|S)$, so the difference does not matter. But when ρ is not zero, it does not. $\text{Var}(P|S)$ is the variance of P given S , so the term $\rho \cdot \sqrt{\text{stuff}} \cdot S$ is a constant and does not contribute to $\text{Var}(P|S)$. Again, $\text{Var}(P|S)$ is not in general a scalar, it's a random variable itself. (It would be clearer to keep track of what RVs the variance is over, e.g. writing $\text{Var}_{G,e_p|S}[P(G,e_p,S)]$ instead of $\text{Var}(P|S)$.) So I don't know what to make of these calculations. But the bigger issue, I think, is that $P|S$ is not what we care about. What we care about is the distribution of P among folks who have developed the

disease, which is $P|S > k$. (Tetrachoric correlations might be useful here?)

Speculation: Given the calculations appear to be on shaky ground, and many of the scenarios considered and highlighted in the discussion are outside the support of the presented data (e.g. a progression phenotype with high heritability or strong genetic correlation between incidence and progression), it seems reasonable to cut these bits and put them in another paper where the maths can be developed a little more carefully and explained more fully. Lines 239 thru 291 and 351 to 362, 384-387 and corresponding figures and supplemental tables could be cut. The point re: index event bias could be made by applying slopehunter to the empirical results.

Reviewer #3:

Remarks to the Author:

Summary of key results

Using international biobanks, the authors looked at 10 diseases and compared genetic associations for disease susceptibility versus progression to cause related death. They found disease susceptibility variants did not robustly associate with progression, and only one significant association for disease progression with HF was found through GWAS (although effect size was in the opposite directions to disease onset).

The authors also used several theoretical approaches to demonstrate that the genetic architecture of disease onset is different to progression and that larger sample sizes or better phenotyping would help identify more relevant genetic associations specific to disease progression.

A PRS of longevity appeared to perform better than the PRS of progression in half of the diseases to predict disease related death, suggesting the chosen phenotype may not be true reflection of disease biological progression but one reflecting processes related to death and longevity. This calls into question whether the study really reflects disease onset vs progression and is discussed below.

Originality and Significance

The work builds on prior observations, as others have documented the lack of overlap between disease onset and progression in genetic studies previously and some of these are referenced in the introduction. This study differs in exploring this issue at scale across multiple diseases using several biobanks. The authors also include theoretical approaches to explore issues like index event bias as explanations for the attenuated associations.

The study confirms what we know, albeit it does it in a systematic way, but in my opinion does not shed any new light on why associations are attenuated or absent or which variants may be important for progression.

Part of the problem is that the choice of the progression phenotype is not truly reflective of the underlying biology of the chosen diseases. Choosing death as the progression endpoint also does not fit with the rationale for study given in the introduction about mechanistic understanding and drug targets or being clinically relevant for secondary prevention.

For example, for CHD, disease onset may be related to factors driving new plaques, and progression would be interpreted as expansion of plaque or plaque rupture, which could be druggable pathways.

Furthermore, for a disease like AF, it does not make sense to define progression as AF-related death. People don't die from AF. They may die from the consequence of AF such as stroke or heart failure, so again the progression phenotype does not really reflect progression. For AF, something like AF recurrence, progression to permanent AF or repeat ablations etc. would be more relevant.

Data & methodology: validity of approach, quality of data, quality of presentation

On the whole the analyses and data presentation are done to a good standard, and I did not have any major concerns.

Perhaps there are too many analyses squeezed into one paper. For example, in just one paragraph, the results of 10 new GWASs of progression, along with the association of numerous disease susceptibility variants with progression, are presented.

It would be helpful to label or number the different analyses or results sections and match these up to the study diagram.

Appropriate use of statistics and treatment of uncertainties

There is an appropriate use of statistics for essential GWAS and other genetic associations including PRS methods.

An expert in the theoretical modeling aspect of the study with appropriate statistical methodology expertise should be

consulted.

It is commendable that the authors have tried to explore the reasons why disease progression analyses did not identify new findings. My concern is that, with a flawed progression phenotype, everything that follows is also fundamentally weakened. It would almost have been better to drill down into one area, one disease with in-depth phenotyping rather than superficial data analysis of multiple diseases.

Conclusions: robustness, validity, reliability

The conclusions are fair. The limitations section acknowledges many of the limitations listed above.

Suggested improvements: experiments, data for possible revision

Looking at true disease progression rather than death would be of most value, although this would require a fundamental re-analysis.

References: appropriate credit to previous work?

Referencing is appropriate

Clarity and context: lucidity of abstract/summary, appropriateness of abstract, introduction, and conclusions

Clearly written but grammatical errors throughout, would warrant careful proofing. I thought the introduction and discussion were written to a good standard and were easy to follow.

Version 1:

Decision Letter:

29th August 2024

Dear Andrea,

I would like to begin by apologizing for the unusually long processing delay. Your revised Article "Limited overlap between genetic effects on disease susceptibility and disease survival" has been seen by two of the original referees. (Reviewer #2 was also asked to comment on the revision but has not submitted a review to date.) You will see from the comments of Reviewers #1 and #3 that, while they find the study improved, they have raised a few ongoing concerns. We remain interested in the possibility of publishing your study in Nature Genetics, but we would like to consider your response to these concerns in the form of a further revision before we make a final decision on publication.

As before, to guide the scope of the revisions, the editors discuss the referee reports in detail within the team, including with the chief editor, with a view to identifying key priorities that should be addressed in revision, and sometimes overruling referee requests that are deemed beyond the scope of the current study. In this case, we ask that you address the concerns regarding the heritability estimates for disease susceptibility reported in Table 2 in relation to previously published estimates, extend the analyses to include the all-cause mortality polygenic score from Meisner et al. 2020, and clarify the rationale for studying microvascular and macrovascular complications as measures of type 2 diabetes progression. We again hope that you will find this prioritized set of referee points to be useful when revising your study. Please do not hesitate to get in touch if you would like to discuss these issues further.

We therefore invite you to revise your manuscript taking into account all reviewer and editor comments. Please highlight all changes in the manuscript text file. At this stage, we will need you to upload a copy of the manuscript in MS Word .docx or similar editable format.

*2) If you have not done so already, please begin to revise your manuscript so that it conforms to our Article format instructions, available

[here](http://www.nature.com/ng/authors/article_types/index.html).
Refer also to any guidelines provided in this letter.

*3) Include a revised version of any required Reporting Summary: <https://www.nature.com/documents/nr-reporting-summary.pdf>

Link Redacted

We hope to receive your revised manuscript within 8 weeks. If you cannot send it within this time, please let us know.

Nature Genetics is committed to improving transparency in authorship. As part of our efforts in this direction, we are now requesting that all authors identified as 'corresponding author' on published papers create and link their Open Researcher and Contributor Identifier (ORCID) with their account on the Manuscript Tracking System (MTS), prior to acceptance. ORCID helps the scientific community achieve unambiguous attribution of all scholarly contributions. You can create and link your ORCID from the home page of the MTS by clicking on 'Modify my Springer Nature account'. For more information, please visit www.springernature.com/orcid.

Sincerely,
Kyle

Kyle Vogan, PhD
Senior Editor
Nature Genetics
<https://orcid.org/0000-0001-9565-9665>

Referee expertise:

Referee #1: Genetics, common diseases, statistical methods

Referee #2: Genetic epidemiology, cancer, statistical methods

Referee #3: Genetics, cardiovascular diseases, clinical translation

Reviewers' Comments:

Reviewer #1:
Remarks to the Author:

I believe the message of the paper is now simpler after removing the abstract mathematical framework introduced in the earlier version. The additional analysis of type-2 diabetes progression is interesting. I still have a couple of residual concerns.

1) I am baffled by the extremely low estimate of heritability of susceptibility reported in Table 2. For many of these diseases, estimates of GWAS heritability are available from other studies and they are much larger in magnitude. For example, for breast cancer, the LD-score observed scale heritability (i.e. heritability in log-OR scale by which summary-stats are typically available) is about 0.50 (see Table 1, Zhang et al., Nature Genetics, 2020, PMID: 32424353). In contrast, the Table 2 in this paper report this heritability to be only 0.01! There is something seriously off here and I am not sure how the authors missed the serious discrepancy with the published literature. Please look into this more carefully and compare your estimates of GWAS heritability estimates for susceptibility with those published literature. And also add more discussion about heritability results given this is a critical piece of data.

2) Further the response to my comment regarding the all-cause mortality PGS (Meisner et al., 2020) is bit disappointing. The authors just cursorily added the reference in the discussion, but actually did not try to use this score to see if this can predict mortality among diagnosed patients. There was no justification why they prefer to use the longevity PGS developed by Timmers et al., 2019, while the Meisner paper clearly showed that one can get better predictor of mortality by considering combining PGS of different diseases and risk-factors. Since the two PGS were developed in different ways, one is based on targeting longevity itself, and the other by combining PGS of known traits associated with mortality, there is a nice opportunity here to see how they comparatively perform to predict mortality conditional on diagnosis.

Reviewer #2:

None

Reviewer #3:

Remarks to the Author:

The authors have revised the manuscript and responded to initial comments, making the paper easier to read, especially with the removal of the theoretical modelling component.

The fundamental challenge of using disease-specific mortality as a phenotype of progression - for biological insights - remains a major problem in the study design, even though acknowledged by the authors.

The new inclusion of type 2 diabetes complications, macrovascular and microvascular as a measure of progression, was not easy to understand. For example, we know that progression to microvascular disease has been proven to be linked to glucose control. Is a diabetes complication due to poor control and affecting the nerves or vessels biologically the same as diabetes (insulin resistance) progression per se? I'm not sure this additional analysis helps to address the challenge of refining the progression phenotype.

Version 2:

Decision Letter:

29th October 2024

Dear Andrea,

Your revised Article "Limited overlap between genetic effects on disease susceptibility and disease survival" has been seen by Reviewer #1, and as you will see from the comments below, Reviewer #1 has requested a few additional revisions and clarifications. We remain interested in the possibility of publishing your study in Nature Genetics, but we would like to see your response to these remaining requests in the form of a revised manuscript before we make a final decision on publication.

When preparing your revision, please highlight all changes in the manuscript text file. At this stage, we will need you to upload a copy of the manuscript in MS Word .docx or similar editable format.

We are committed to providing a fair and constructive peer-review process. Do not hesitate to contact us if there are specific requests that you believe are technically impossible or unlikely to yield a meaningful outcome.

*1) Include a "Response to referee" document detailing, point-by-point, how you addressed each referee comment. If no action was taken to address a point, you must provide a compelling argument. This response may be sent back to the referee along with the revised manuscript.

*2) If you have not done so already, please begin to revise your manuscript so that it conforms to our Article format instructions, available

[here](http://www.nature.com/ng/authors/article_types/index.html).

*3) Include a revised version of any required Reporting Summary (<https://www.nature.com/documents/nr-reporting-summary.pdf>). It will be available to referees (and, potentially, statisticians) to aid in their evaluation if the manuscript goes back for peer review. A revised checklist is essential for re-review of the paper.

Please be aware of our [guidelines](https://www.nature.com/nature-research/editorial-policies/image-integrity) on digital image standards.

Link Redacted

We hope to receive your revised manuscript within four weeks. If you cannot send it within this time, please let us know.

Nature Genetics is committed to improving transparency in authorship. As part of our efforts in this direction, we are now requesting that all authors identified as 'corresponding author' on published papers create and link their Open Researcher and Contributor Identifier (ORCID) with their account on the Manuscript Tracking System (MTS), prior to acceptance. ORCID helps the scientific community achieve unambiguous attribution of all scholarly contributions. You can create and link your ORCID from the home page of the MTS by clicking on 'Modify my Springer Nature account'. For more information, please visit www.springernature.com/orcid.

Sincerely,
Kyle

Kyle Vogan, PhD
Senior Editor
Nature Genetics
<https://orcid.org/0000-0001-9565-9665>

Referee expertise:

Referee #1: Genetics, common diseases, statistical methods

Reviewers' Comments:

Reviewer #1 (Remarks to the Author):

The authors have further clarified some of the issues. I have one modest recommendation and one minor comment.

1) Since the authors note that low estimates of heritability for disease risk may be due to problems with bias in LD-score regression in small samples, I would suggest that the authors also report in the same table either the LD-score estimates of heritability from the full sample or GCTA-based heritability on downsample. As estimate of heritability is not supposed to depend on sample size as much, it may confuse readers why the heritability of risk is so low for disease risks.

2) For a number of diseases, e.g. stroke and CAD, the composite PGS seems to show much stronger association with disease-specific mortality. Are the estimates of HR being presented in the standardized scale (by sd unit of the respective PGS)? Also, the confidence interval for the HR associated with composite PGS is much wider than that based on lifespan PGS. Any explanation? The authors state the results are "similar" between the two types of PGS, but there are some notable differences and so I would clarify the conclusion a bit more.

Version 3:

Decision Letter:

Our ref: NG-A63608R2

4th April 2025

Dear Andrea,

Thank you for submitting your revised manuscript "Limited overlap between genetic effects on disease susceptibility and disease survival" (NG-A63608R2). In light of your responses to Reviewer #1, we will be happy in principle to publish your study in Nature Genetics as an Article pending final revisions to comply with our editorial and formatting guidelines.

We are now performing detailed checks on your paper, and we will send you a checklist detailing our editorial and formatting requirements soon. Please do not upload the final materials or make any revisions until you receive this additional

information from us.

Thank you again for your interest in Nature Genetics. Please do not hesitate to contact me if you have any questions.

Sincerely,
Kyle

Kyle Vogan, PhD
Senior Editor
Nature Genetics
<https://orcid.org/0000-0001-9565-9665>

Overall comments from authors

We would like to thank the editor and reviewers' time and comments on our manuscript. Based on the feedback received, we have undertaken three major changes in the manuscript. We hope this has made the results clearer and more robust.

First, we have completely removed the theoretical and simulation section from this manuscript, which was criticized by several reviewers as being too abstract and unsupported by empirical data. We have only kept simulations concerning index event bias.

Second, we addressed the main criticism that disease-specific mortality is not a relevant measure of progression for some of the diseases included by considering alternative, clinically relevant, definitions of progression. Specifically, we focused on microvascular and macrovascular complications of type 2 diabetes. We have also removed atrial fibrillation because disease-specific mortality was indeed a poor proxy for progression.

Third, we have rerun all analyses using the most recent FinnGen data release and updated results accordingly, thus increasing the patient number of the largest disease from 99,666 to 124,682 individuals.

Our conclusions did not change, but the additional analyses on progression of type 2 diabetes allows to further expand our conclusions on how genetic signals discovered in the general population, can directly inform progression among diseased individuals. This is exemplified by the discovery of new signals for diabetic macrovascular complications, which overlap with well-known signals for cardiovascular diseases in the general population.

Reviewer #1

Remarks to the Author:

The article presents results from GWAS of disease-specific mortality for common complex diseases. This is the largest study of its kind, and the main message is that there is very little signal of genetic effects on disease progression among diagnosed patients. This is mostly consistent with other disease-specific efforts. This mostly null study points to challenges associated with finding genetic determinants of disease progression using the GWAS approach. The authors spend considerable time developing an abstract (unobservable) model that considers disease progression in the general population. While the math is correct and interesting, I was not convinced of the practical value of such thinking. Below are some specific comments.

Comment 1: The authors should provide estimates of heritability for disease-specific mortality and susceptibility across the different diseases side by side in a main table. Such a table will help readers understand the impact of genetic contribution on these two traits. It would be further useful if, in the table legend, the authors could remark on the relationship of these definitions of heritability, which are directly estimable, to the concepts of heritability defined in their mathematical framework. Currently, the results for disease-specific mortality are apparently presented in the Supplemental Table (S2), but I could not dig out those results because of the huge size of S1.

Answer: Thank you for your suggestion. We agree that it is an important information and should be of high interest to the audience. We have added observed heritability in **Table 2**, where we compare the genetics of disease mortality and susceptibility measured in FinnGen and UK biobank under the same sample size.

Comment 2: The authors seem to have missed a key reference to a prior study on polygenic modeling of all-cause mortality (longevity) using combinations of susceptibility genetic scores (Meisner et al., AJHG, 2020, PMID: 32758451). In fact, this defined a more powerful PGS for predicting longevity than the PGS that has been defined directly based on GWAS of longevity phenotype (Timmers et al., 2019). Meisner et al also reported the association of the susceptibility PRS with disease-specific mortality across the different diseases. All of these are very relevant for this paper.

Answer: Thank you for pointing this out. We have now added this reference at page 12, line 323-324.

Comment 3: Line 132-136. I am surprised by the hypothesis here that one expects the effects of SNPs to be the same on the disease risk vs disease progression. By default, I would expect they will not be the same. The more relevant hypothesis is that whether the susceptibility alleles have some kind of effect on progression and in what direction. I did not find Figure 2, which investigates the hypothesis of equal effect, to be interesting, and it should be moved to supplement.

Answer: We agree that equal effect is a strong hypothesis. We have now revised **Figure 2** correspondingly. We have removed the Bayesian approach to assign the genetic variants to three groups (only disease-specific mortality, only susceptibility and equal effect) and instead simply reported the effects for disease-specific mortality vs susceptibility in a scatter plot. This allows to highlight the

relationship (or lack thereof) between the effects sizes from the two analyses without imposing unrealistic assumptions of equal effects.

Figure 2. Relationship between variant effects (one for each locus) on disease susceptibility (x-axis) and disease-specific mortality (y-axis). Variants were selected either because genome-wide significance for susceptibility in the largest disease specific GWAS or because genome-wide significance for disease-specific mortality in the current study, indicated by color of the dot (red: significant in susceptibility GWAS; blue: significant in mortality GWAS). Only one locus for heart failure mortality was genome-wide significant.

Comment 3: The authors spend considerable effort developing an abstract mathematical model for disease progression in the general population, including individuals not diagnosed with the disease. The point being that ideally, one would like to estimate the effects of genetic variants on the disease progression in that abstract setting. I was not very convinced with this argument. The authors did not explain or demonstrate what is the practical utility of such models for obtaining insight into biology or, risk prediction, or both. Second, such a model cannot be fitted based on the observable data, so everything stays very hypothetical. Even in their own discussion (lines 319-323) regarding the cause for low heritability for disease progression, the authors argued for potential effects of heterogeneous treatments that are only relevant for the population of affected individuals. I do think the math is correct and interesting, but I failed to see a lot of practical value in it.

The authors mostly used simulation studies to examine the relationship between “idealized” and “observable” genetic effects on disease progression. I wondered if they could have used the mathematical results from the real datasets to say more about the parameters of the idealized model for disease progression in the general population. The only direct relevance of all the math in the interpretation of the results from real studies I could see is the investigation of index-event bias.

The supplementary methods section is unnecessarily detailed in places and can be shortened substantially. For example, the theoretical results must only be derived once assuming a general value of ρ , and then the formula for $\rho=0$ case can be derived as a special case. What is the point of introducing the problem through a general function $f(S)$ where all of the calculations actually require $f(S)$ to be linear? Also, there was a mention of β being sparse, but that is never needed in the calculations.

Answer: following the comments from the other reviewers and the editor we have removed the majority of the theoretical and simulation results, including figure 5 and the corresponding section. We have only kept simulations concerning index event bias.

Reviewer #2:

Remarks to the Author:

Comment 1: This paper explores the genetic contribution to disease-specific survival across ten common diseases using a relatively large number of cases from seven biobanks. It finds little evidence for strong genetic association with disease-specific survival, suggesting that the strength of association between common variants and disease-specific survival is much weaker than that for common variants and disease incidence. (The paper includes a nice empirical demonstration that, in equivalent sample sizes, GWAS of disease incidence find many more associations than GWAS of disease progression.) The paper also looks at the association between polygenic scores for disease incidence and disease-specific mortality, finding at most modest associations. Finally, the paper proposes a mathematical model for the co-contribution of genetics to incidence and survival and runs simulations based on that model to draw inference about study designs best suited for the study of disease-specific survival, the likelihood and magnitude of index-event bias, etc.

Strengths: Although the individual disease sample sizes are not always as large as previously published studies of disease-specific survival (c.f. breast cancer), they are each relatively large and large in aggregate. Including ten diseases in one paper is nice, as it allows general conclusions and contrasts to be made. For example, it is interesting to see the direction of the slopes of the regression of survival effects on incidence effects vary from disease to disease in Figure 2 and to see the incidence PGS-survival associations vary across disease. (One who only knows the breast and prostate cancer genetic epidemiology literature might be tempted to rashly conclude there is a general inverse relationship between susceptibility and survival--Figures 2 and 3 will set them aright.)

Weaknesses: The "theoretical framework" sections are unclear, speculative, and, I suspect, contain mathematical errors.

Clarity: many readers will not catch the fact that the additive liability model introduced in the theoretical framework section does not have a clear relationship to the empirical results presented from proportional hazards regressions. (The "heritability" of a censored survival trait is rather tricky to define; there's not a natural way to partition variance into "genetic" and "environmental" components under a proportional hazards model.) The paper flags this difference near the end of the discussion, 100 lines later. Nor does the paper ever, anywhere that I have noticed, define what is meant by liability. One presumes based on tradition that there is a probit link relating probabilities of disease incidence and survival to the additive liability. Would be nice to see that explicitly stated.

Mathematical concerns: The theoretical framework section needs another close read for mathematical rigor. I had several concerns. First, the liability of within-patient disease progression is not $P|S$ or $P|S=s$, it is $P|S>k$, where k is the liability threshold for disease incidence. $P|S>k$ seems to be the latent phenotype of interest here, not $P|S=s$. Second, the notation $\text{Var}(P|S)$, $G_{P|S}$ etc. seems rather loose, and may lead some calculations astray. In standard prob/stats notation, $\text{Var}(P|S)$ is a random variable; specifically, it is a function of the random variable S . It is not a fixed quantity. In the $\rho=0$ case, the S drops out of $\text{Var}(P|S)$, so the difference does not matter. But when ρ is not zero, it does not. $\text{Var}(P|S)$ is the variance of P given S , so the term $\rho \cdot \sqrt{\text{stuff}} \cdot S$ is a constant and does not contribute to $\text{Var}(P|S)$. Again, $\text{Var}(P|S)$ is not in general a scalar, it's a random variable itself. (It would be clearer to keep track of what RVs the variance is over, e.g. writing $\text{Var}_{\{G,e_p|S\}}[P(G,e_p,S)]$ instead of $\text{Var}(P|S)$.) So I don't know what to make of these calculations. But the bigger issue, I think, is that $P|S$ is not what we care about. What we care about is the distribution of P among folks who have developed the disease, which is $P|S>k$. (Tetrachoric correlations might be useful here?)

Speculation: Given the calculations appear to be on shaky ground, and many of the scenarios considered and highlighted in the discussion are outside the support of the presented data (e.g. a progression phenotype with high heritability or strong genetic correlation between incidence and progression), it seems reasonable to cut these bits and put them in another paper where the maths can be developed a little more carefully and explained more fully. Lines 239 thru 291 and 351 to 362, 384-387 and corresponding figures and supplemental tables could be cut. The point re: index event bias could be made by applying slopehunter to the empirical results.

Answer: Thank you for the detailed feedback. We have received similar suggestion from other reviewers and the editor and we have removed the majority of the theoretical and simulation results, including figure 5 and the corresponding section. We decided to keep the simulations concerning index event bias since we believe that only with a known underlying variant effect can the impact of this bias under various genetic architecture be examined, so is the effectiveness of slopehunter-like correction. We kept the math to a minimum level and made it consistent to the original slopehunter theoretical framework. We have added in the main text at page 11, line 279 a specific section about index-event bias:

Potential role of index event bias

Noticing an attenuation of variant effects in patients' progression GWAS under the same sample sizes, we suspected index event bias can play a role. Therefore, we carried out simulation under a simple liability threshold model and explored the impact of index event bias by introducing a shared non-genetic risk factor accounting for various proportions of the liability in disease susceptibility and progression. We compared the simulated effect of causal genetic variants on progression with the observed effect from the progression

GWAS and found larger differences when the shared non-genetic component accounted for higher liability variance, indicating higher impact of index event bias (**Figure S19**). A correction approach similar to slope-hunter (Mahmoud et al., 2022) reduced the bias improving the concordance with the true simulated effects. However, in the scenario of a low progression heritability, which is consistent with our empirical findings for disease mortality, index event bias correction showed a limited impact as we observed no genetic variants significantly associated with disease progression before or after bias correction (**Table S19**).

Reviewer #3:

Remarks to the Author:

Using international biobanks, the authors looked at 10 diseases and compared genetic associations for disease susceptibility versus progression to cause related death. They found disease susceptibility variants did not robustly associate with progression, and only one significant association for disease progression with HF was found through GWAS (although effect size was in the opposite directions to disease onset). The authors also used several theoretical approaches to demonstrate that the genetic architecture of disease onset is different to progression and that larger sample sizes or better phenotyping would help identify more relevant genetic associations specific to disease progression.

A PRS of longevity appeared to perform better than the PRS of progression in half of the diseases to predict disease related death, suggesting the chosen phenotype may not be true reflection of disease biological progression but one reflecting processes related to death and longevity. This calls into question whether the study really reflects disease onset vs progression and is discussed below.

Comment 1: The work builds on prior observations, as others have documented the lack of overlap between disease onset and progression in genetic studies previously and some of these are referenced in the introduction. This study differs in exploring this issue at scale across multiple diseases using several biobanks. The authors also include theoretical approaches to explore issues like index event bias as explanations for the attenuated associations.

The study confirms what we know, albeit it does it in a systematic way, but in my opinion does not shed any new light on why associations are attenuated or absent or which variants may be important for progression.

Part of the problem is that the choice of the progression phenotype is not truly reflective of the underlying biology of the chosen diseases. Choosing death as the progression endpoint also does not fit with the rationale for study given in the introduction about mechanistic understanding and drug targets or being clinically relevant for secondary prevention.

For example, for CHD, disease onset may be related to factors driving new plaques, and progression would be interpreted as expansion of plaque or plaque rupture, which could be druggable pathways.

Furthermore, for a disease like AF, it does not make sense to define progression as AF-related death. People don't die from AF. They may die from the consequence of AF such as stroke or heart failure, so again the progression phenotype does not really reflect progression. For AF, something like AF recurrence, progression to permanent AF or repeat ablations etc. would be more relevant.

Answer: *Thank you for the feedback. We agree that disease-specific mortality might not be the best proxy for the progression for several of the disease considered. Our approach is based on the realization that*

many of the more detailed phenotypes required to study disease progression (e.g. plaque rupture in the CHD example the Reviewer makes) will not be available in large-scale biobanks and disease-specific mortality might represent a good proxy. Whether this is the case, it would depend on the specific disease. For example, cancer survival is a well-established metric of progression. Part of this work is about showing that disease-specific mortality might indeed not be a good proxy of disease progression. Nonetheless, we agree, in spirit, with the Reviewer and have addressed the concerns of non-adequacy of disease-specific mortality as proxy of disease progression in two ways.

First, we have removed atrial fibrillation from the analyses, as we agree it does not make sense to use death as proxy for atrial fibrillation progression. After consulting with our clinical collaborators, we concluded that it is hard to find a good alternative progression definition due to the inaccurate time this disease is recorded in the electronic health records. In practice, by the time patients received a diagnosis of atrial fibrillation, many already have other undiagnosed heart problems.

Second, we have focused on other progression definitions for type 2 diabetes and conducted additional analyses in FinnGen and UK biobanks. We choose type 2 diabetes as use case because progression, defined as macro- and microvascular complications, can be easily captured in the electronic health records. This new analysis has led to the discovery of three new loci for type 2 diabetes progression and supported the idea that genetic signals for disease progression can be discovered by using phenotypes measured in the general population but related to the progression phenotypes. In the case of the significant loci for type 2 diabetes macro- and microvascular complications, they could be similarly detected by using related phenotype definitions among people without type 2 diabetes.

To expand the progression definitions beyond disease-specific mortality we have added a new section describing the results for type 2 diabetes, including a new **figure 5** and new supplementary figures **S17, S18**.

Figure 5. GWAS of type 2 diabetes progression defined as macrovascular complications. The upper Manhattan plots displays the results of a GWAS carried out in individuals with type 2 diabetes. The lower Manhattan plots display the results of GWAS carried out in individuals without type 2 diabetes where proxies phenotypes for macrovascular complications were used instead. Two GWAS were matched to the same number of cases and controls to guarantee similar power.

Figure S17. Manhattan plot for meta-analyzed type 2 diabetic microvascular complication GWAS.

Figure S18. Association between various relevant PGS and type 2 diabetic complications. Horizontal solid lines represent 95% CI. Also see Table S5 for quantitative results

We have also extensively discussed these results in the discussion

Page 12, line 327-331

However, a definition that maximize data availability might not be one that best reflects the genetic etiology of a specific disease. As a proof of concept, we considered two clinically relevant definitions of type 2 diabetes progression that could easily be captured in the electronic health records. As such, we were able to identify one genome-wide significant locus that was not detected when considering mortality.

Page 12, line 353-364

Similarly, the significant findings for macrovascular complications in type 2 diabetes were recapitulated in the general population when looking at related phenotypes, even among individuals without type 2 diabetes. The observed genetic overlap opens the possibility of shared underlying biology between these two diseases. Importantly PGSs for related traits in the general population were better predictor of type 2 diabetes progression than a PGS for type 2 diabetes susceptibility. Methods for cross-trait PGS (Kember et al., 2021) might be leveraged to obtain progression PGS based on existing GWAS results in the general population. This relationship is however not always obvious. For diabetic microvascular complications, it is not as straightforward to find any population equivalent measurement. On the other hand, this could indicate that such progression definition is more unique to the diseased cohort and have the potential in yielding truly progression specific genetic signals if power permits.

And mentioned the results in the abstract

We explored alternative definitions of disease progression and show that genetic signals for macrovascular complication in type 2 diabetes can be identified using related phenotypes in the general population.

Comment 2: On the whole the analyses and data presentation are done to a good standard, and I did not have any major concerns.

Perhaps there are too many analyses squeezed into one paper. For example, in just one paragraph, the results of 10 new GWASs of progression, along with the association of numerous disease susceptibility variants with progression, are presented.

It would be helpful to label or number the different analyses or results sections and match these up to the study diagram.

Answer: Thank you for the suggestion. We have updated our figure 1 to include an overview of all analyses in the study and purpose corresponding to each analysis. Hope this could make it easier to follow.

Figure 1. In this study, using data from seven biobanks, we investigated the genetic similarity between disease susceptibility and disease progression, defined as disease-specific mortality. We selected nine diseases and ran GWASs of disease-specific mortality among disease individuals. We then compared the genetic architecture of disease susceptibility and mortality focusing on both single variant and aggregated polygenic effects. We further explored the impact of alternative progression definitions and theoretical impact of index event bias on the results.

Comment 3: There is an appropriate use of statistics for essential GWAS and other genetic associations including PRS methods. An expert in the theoretical modeling aspect of the study with appropriate statistical methodology expertise should be consulted. It is commendable that the authors have tried to explore the reasons why disease progression analyses did not identify new findings. My concern is that, with a flawed progression phenotype, everything that follows is also fundamentally weakened. It would

almost have been better to drill down into one area, one disease with in-depth phenotyping rather than superficial data analysis of multiple diseases.

The conclusions are fair. The limitations section acknowledges many of the limitations listed above. Looking at true disease progression rather than death would be of most value, although this would require a fundamental re-analysis. Referencing is appropriate. Clearly written but grammatical errors throughout, would warrant careful proofing. I thought the introduction and discussion were written to a good standard and were easy to follow.

***Answer:** We refer to answer to comment 1*

Overall comments from authors

We would like to thank the editor and reviewers' time and comments on our manuscript. Based on the feedback received, we carried out suggested experiment and added clarifications in the manuscript. We hope this has made the results clearer and more robust.

Reviewers' Comments:

Reviewer #1:

Remarks to the Author:

I believe the message of the paper is now simpler after removing the abstract mathematical framework introduced in the earlier version. The additional analysis of type-2 diabetes progression is interesting. I still have a couple of residual concerns.

1) I am baffled by the extremely low estimate of heritability of susceptibility reported in Table 2. For many of these diseases, estimates of GWAS heritability are available from other studies and they are much larger in magnitude. For example, for breast cancer, the LD-score observed scale heritability (i.e. heritability in log-OR scale by which summary-stats are typically available) is about 0.50 (see Table 1, Zhang et al., Nature Genetics, 2020, PMID: 32424353). In contrast, the Table 2 in this paper report this heritability to be only 0.01! There is something seriously off here and I am not sure how the authors missed the serious discrepancy with the published literature. Please look into this more carefully and compare your estimates of GWAS heritability estimates for susceptibility with those published literature. And also add more discussion about heritability results given this is a critical piece of data.

Answer: Thank you for pointing that out. Please note that Table 2 presents result from down-sampled GWAS analyses, where we deliberately under sampled the diseased cohort to match the exact sample size we used in our mortality GWAS. The goal was to demonstrate that lack of genetic signals in the mortality GWAS could not be simply attributed to a smaller sample size, but also, the overall weaker strength of signals in disease mortality.

It is known (<http://www.nealelab.is/blog/2017/9/20/insights-from-estimates-of-snp-heritability-for-2000-traits-and-disorders-in-uk-biobank>) that LD score regression underestimate heritability when the GWAS is not well powered, as in this case. This is the main reason why our heritability results are smaller than what previously shown.

In addition, we were surprised about the large heritability reported in the breast cancer GWAS mentioned by the Reviewer (Zhang et al., 2020). Unfortunately, the summary statistics for that paper are not available to download anymore. But we did check the LDSC heritability for the breast cancer GWAS we used to develop PGS (Michailidou et al., 2017 - 122,977

cases and 105,974 controls) and the LDSC heritability estimate on an observed scale was 0.133, which is closer to what we report.

One reason why the heritability of Zhang et al., 2020 is substantially higher than what reported by Michailidou et al., 2017 is that Zhang et al., 2020 report heritability on a frailty-scale, which results in higher estimates than observed-scale heritabilities.

In conclusion two factors might explain the smaller heritability:

- 1) Bias in LD score regression estimate due to small sample size caused by the down-sample experiment
- 2) Heritability reported on observed rather than frailty scale

2) Further the response to my comment regarding the all-cause mortality PGS (Meisner et al., 2020) is bit disappointing. The authors just cursorily added the reference in the discussion, but actually did not try to use this score to see if this can predict mortality among diagnosed patients. There was no justification why they prefer to use the longevity PGS developed by Timmers et al., 2019, while the Meisner paper clearly showed that one can get better predictor of mortality by considering combining PGS of different diseases and risk-factors. Since the two PGS were developed in different ways, one is based on targeting longevity itself, and the other by combining PGS of known traits associated with mortality, there is a nice opportunity here to see how they comparatively perform to predict mortality conditional on diagnosis.

Answer: Thank you for the suggestion. We chose the longevity PGS in the main text for its interpretability, as it can be explained as the “genetics of mortality”. We agree that it would also be interesting to see how that compares to a PGS developed to predict mortality using a combination of genetics of multiple risk factors. We have added this experiment in the supplement for comparison and the text below on page 9, line 232-234:

Another composite mortality PGS developed from genetics of multiple risk factors (Meisner et al., 2020) showed similar performances and was significantly associated with disease-specific mortality for four out of nine diseases (Figure S16, Table S11).

Figure 16. Association between a PGS for disease susceptibility (orange dots), longevity (dark blue dots) and composite mortality PGS (light blue dots) with disease-specific mortality in FinnGen. Disease susceptibility PGSs were derived from published large-scale GWAS for each disease. Longevity PGS was derived from (Timmers et al., 2019). Composite mortality PGS was derived from (Meisner et al., 2020). Horizontal solid lines represent 95% CI. Also see Table S11 for quantitative results.

Reviewer #3:

Remarks to the Author:

The authors have revised the manuscript and responded to initial comments, making the paper easier to read, especially with the removal of the theoretical modelling component.

The fundamental challenge of using disease-specific mortality as a phenotype of

progression - for biological insights - remains a major problem in the study design, even though acknowledged by the authors.

The new inclusion of type 2 diabetes complications, macrovascular and microvascular as a measure of progression, was not easy to understand. For example, we know that progression to microvascular disease has been proven to be linked to glucose control. Is a diabetes complication due to poor control and affecting the nerves or vessels biologically the same as diabetes (insulin resistance) progression per se? I'm not sure this additional analysis helps to address the challenge of refining the progression phenotype.

Answer: Thank you for the insight. In this case, our motivation was more of to understand the utility of using genetic, e.g. GWAS variants or PGS, to predict patient's prognostic outcomes, rather than to gain biological insights. However, we do agree that one of the reasons that it does not work, as pointed out by the reviewer, could be due to the difference in biological mechanisms underlying disease progression. We have added this into the discussion at page 12, line 345-347.

Lastly, even with all three aspects taken into account, there is still the possibility that we do not observe comparable genetic effects on disease susceptibility and progression, simply due to the reason that their underlying biological mechanisms are truly distinct.

Overall comments from authors

We would like to thank the editor and reviewers' time and comments on our manuscript. Based on the feedback received, we carried out experiment and revise the text. We hope could clarify the message we would like to convey through this manuscript.

Reviewers' Comments:

- 1) Since the authors note that low estimates of heritability for disease risk may be due to problems with bias in LD-score regression in small samples, I would suggest that the authors also report in the same table either the LD-score estimates of heritability from the full sample or GCTA-based heritability on downsample. As estimate of heritability is not supposed to depend on sample size as much, it may confuse readers why the heritability of risk is so low for disease risks.

Thank you for the suggestion. We have now calculated ldsc heritability from the complete FinnGen and UK biobank meta-analyzed GWAS and reported these results in supplementary table 2. Moreover, given the challenges in interpreting heritability estimates at small sample sizes, as also pointed out by the Reviewer, we have decided to move all the heritability estimates in supplementary table 2. We think this will make Table 2 more clear.

- 2) For a number of diseases, e.g. stroke and CAD, the composite PGS seems to show much stronger association with disease-specific mortality. Are the estimates of HR being presented in the standardized scale (by sd unit of the respective PGS)? Also, the confidence interval for the HR associated with composite PGS is much wider than that based on lifespan PGS. Any explanation? The authors state the results are "similar" between the two types of PGS, but there are some notable differences and so I would clarify the conclusion a bit more.

Thank you for the careful reading. Yes, HRs are presented in the standardized scale. And the reason that composite PGS has a larger confidence interval is that weights for the composite PGS are provided for male and female separately. Therefore, the score was computed separately by sex, and then combined during the regression for a population effect estimate. The large confidence interval in this case was resulted from a difference in PGS distribution between males and females. We have added text in the following text in method section on line 490-492:

For the composite mortality PGS, we used sex-stratified SNP weights developed by (Meisner et al., 2020). Scores for males and females were computed separately, and subsequently combined during the association step for a population effect estimate.

We have also clarified this in Suppl Figure 16 legend.

Regarding the statement that results were “similar”, we apologize for the confusion. We intended to mean that the effect of composite PGS and longevity PGS are similar, in comparison to a disease susceptibility PGS – they are both not worse, or better than a susceptibility PGS for predicting disease mortality in most of the cases. We have updated text in the result section at line 230-233 to make this more clear.

We also tested another composite mortality PGS developed from genetics of multiple risk factors (Meisner et al., 2020) in FinnGen. The composite mortality PGS was significantly associated with disease-specific mortality for four out of nine diseases, and outperformed longevity PGS in predicting Type 2 diabetes and coronary artery disease mortality (Figure S16, Table S11).